# The Illusion of Progress? A Critical Look at Test-Time Adaptation for Vision-Language Models

**Lijun Sheng**[1,2], **Jian Liang**[2,3*], **Ran He**[2,3], **Zilei Wang**[1], **Tieniu Tan**[2,3,4]

[1] University of Science and Technology of China
[2] NLPR & MAIS, Institute of Automation, Chinese Academy of Sciences
[3] School of Artificial Intelligence, University of Chinese Academy of Sciences
[4] Nanjing University
slj0728@mail.ustc.edu.cn, liangjian92@gmail.com

## Abstract

Test-time adaptation (TTA) methods have gained significant attention for enhancing the performance of vision-language models (VLMs) such as CLIP during inference, without requiring additional labeled data. However, current TTA researches generally suffer from major limitations such as duplication of baseline results, limited evaluation metrics, inconsistent experimental settings, and insufficient analysis. These problems hinder fair comparisons between TTA methods and make it difficult to assess their practical strengths and weaknesses. To address these challenges, we introduce **TTA-VLM**, a comprehensive benchmark for evaluating TTA methods on VLMs. Our benchmark implements 8 episodic TTA and 7 online TTA methods within a unified and reproducible framework, and evaluates them across 15 widely used datasets. Unlike prior studies focused solely on CLIP, we extend the evaluation to SigLIP—a model trained with a Sigmoid loss—and include training-time tuning methods such as CoOp, MaPLe, and TeCoA to assess generality. Beyond classification accuracy, TTA-VLM incorporates various evaluation metrics, including robustness, calibration, out-of-distribution detection, and stability, enabling a more holistic assessment of TTA methods. Through extensive experiments, we find that 1) existing TTA methods produce limited gains compared to the previous pioneering work; 2) current TTA methods exhibit poor collaboration with training-time fine-tuning methods; 3) accuracy gains frequently come at the cost of reduced model trustworthiness. We release TTA-VLM to provide fair comparison and comprehensive evaluation of TTA methods for VLMs, and we hope it encourages the community to develop more reliable and generalizable TTA strategies. The code is available in https://github.com/TomSheng21/tta-vlm.

## 1 Introduction

Vision-language models (VLMs) [1–5], such as CLIP [1], learn to align visual and textual representations in a shared embedding space, enabling effective performance on a wide range of multi-modal tasks. These models achieve remarkable performance in tasks such as image classification [1, 6, 7], multi-modal retrieval [8–10], and semantic segmentation [11–14]. Especially, the introduction of natural language makes VLMs have better flexibility and generalization than traditional visual classification models. To further enhance the performance of VLMs, numerous training-time approaches [6, 15–18] have been proposed, introducing effective optimization solutions and inference paradigms.

In contrast to training-time approaches, test-time adaptation (TTA) methods [19–23] have attracted increasing attention for their ability to enhance the performance of VLMs during inference without

---

[*]To whom correspondence should be addressed.

requiring additional annotated data. Depending on the type of test data, TTA methods can be broadly categorized into episodic [21, 24, 25] and online [26–29] TTA strategies. Episodic TTA adapts the model to a single test sample, exploring the information inside the sample to improve the model's prediction. Online TTA aims to process a stream of test data, utilizing historical knowledge in the previously seen data to update model behavior.

However, existing TTA research for VLMs suffers from various limitations in experimental setup and analysis. First, most works rely on reported baseline results without reproducing them under consistent experimental settings. This always leads to unfair comparisons due to variations in pre-trained model checkpoints, text prompts, and evaluation protocols. Furthermore, except for a few works [24, 30, 25, 31] that explore alternative evaluation metrics, the majority of research focuses narrowly on accuracy, resulting in an incomplete understanding. Additionally, there is a lack of a systematic evaluation of TTA methods on training-time tuned models [6, 32] and on VLM architectures beyond CLIP, limiting the generalizability of existing conclusions.

To address the limitations of current TTA research for VLMs, we introduce TTA-VLM, a unified benchmark framework for systematic evaluation and fair comparison. TTA-VLM contains 8 episodic and 7 online TTA methods, and evaluates them across 15 widely-used datasets commonly adopted in VLM fine-tuning. Beyond CLIP, we also incorporate training-time methods (i.e., CoOp [6], MaPLe [32], TeCoA [33]) and SigLIP [4] to assess the generality and compatibility of TTA methods with diverse VLMs variants. Our evaluation framework goes beyond accuracy by integrating multiple axes of model behavior: calibration [34], robustness [35], out-of-distribution (OOD) detection performance [36], and stability against abnormal data. We systematically analyze the impact of TTA algorithms on the trustworthiness of VLMs while improving accuracy, so as to give the practitioner a deep understanding. Through extensive experimentation, we obtain several key findings:

- Existing TTA methods yield limited performance gains over the previous pioneering work.
- TTA methods show disappointing collaboration with training-time fine-tuning methods.
- While improving accuracy, TTA methods compromise the trustworthiness of VLMs.

## 2 A Comprehensive Benchmark for Test-Time Adaptation

In this section, we introduce the tasks and methods in the TTA-VLM benchmark. Sec.2.1 defines the paradigms of test-time adaptation on VLMs. Sec.2.2 describes the adaptation methods included in our benchmark. And Sec.2.3 provides information about the pre-trained models and datasets used in our experiments. An overview of the TTA-VLM is provided in Figure 1.

The experimental results are presented in the subsequent sections: In Sec.3, we conduct a fair comparison of TTA methods on both CLIP and SigLIP models. In Sec.4, we explore how well TTA methods collaborate with training-time fine-tuning approaches. And in Sec.5, we analyze the impact of TTA on model trustworthiness beyond classification accuracy.

### 2.1 Test-Time Adaptation Paradigms

**Definition 1 (Episodic Test-Time Adaptation).** Given a pre-trained VLM classifier $f(x)$ and a single test sample $x_{test}$, episodic test-time adaptation refers to the process of adapting $f(x)$ at inference time by using the information in $x_{test}$ with the goal of improving the prediction quality for $x_{test}$.

**Definition 2 (Online Test-Time Adaptation).** Given a pre-trained VLM classifier $f(x)$ and a sequence of test batches $\{\mathcal{B}_{test}^t\}_{t=1,2,...}$, online test-time adaptation refers to the process of adapting $f(x)$ at inference time by leveraging the knowledge inside $f(x)$ and the information in the previous batches during inference time with the goal of improving the prediction quality for current test batch.

### 2.2 Test-Time Adaptation Methods

Our benchmark contains 8 episodic TTA methods and 7 online TTA methods. We provide a brief introduction here; a detailed description and the introduction of TTA methods for other tasks (e.g., segmentation) can be found in the supplementary material.

First, we introduce the episodic methods that adapt to a single test sample. Since a single instance contains limited information, episodic adaptation methods employ Augmix [37] to obtain a batch of

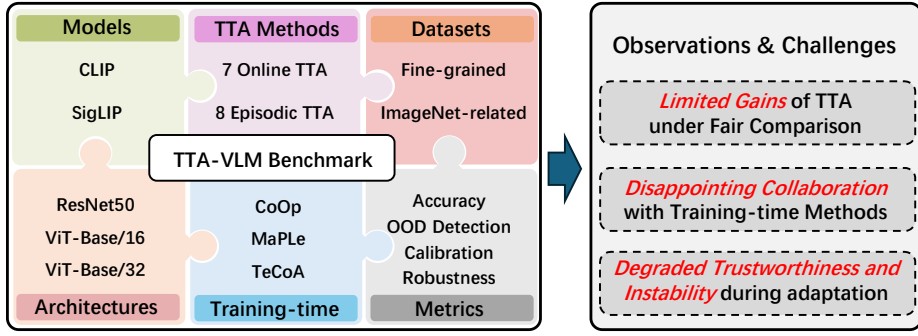

Figure 1: Overview structure of proposed benchmark TTA-VLM.

augmented views. As a pioneering work, **TPT** [21] selects the low-entropy views in the augmented batch and optimizes the textual prompt to minimize their marginal entropy [38]. **C-TPT** [24] introduces a new loss to minimize the dispersion of text features to seek both high accuracy and low calibration error. **RLCF** [39] employs CLIP as a reward model and utilizes reinforcement learning to update the prediction step by step. **MTA** [40] deploys robust MeanShift to calculate the weighting coefficient for ensembling the augmented image features to obtain a better feature. **ZERO** [41] argues that marginal entropy minimization has a minor impact on the average prediction and chooses to directly conduct a hard vote on low-entropy views. **TTL** [42] proposes to optimize the LoRA module inside the image encoder to minimize a designed weighted entropy objective. **TPS** [43] uses the same loss function as TPT and chooses to optimize the shift of text features to combine with multiple text templates. **R-TPT** [25] modifies the objective to pointwise entropy and introduces a reliablility-based ensembling strategy to improve the accuracy and the robustness against adversarial examples.

Then, we provide information about online methods that process streaming test data. **TDA** [26] saves previous high-quality samples into positive and negative caches to improve the prediction of VLMs. **DMN** [44] preserves historical test features and reads out new classifier in an online manner to enhance model performance. **OnZeta** [45] traces the distribution of assigned labels and proposes an online vision proxy learning method to obtain accurate class visual proxies. **BoostAdapter** [46] produces final predictions with the help of boosting samples selected from the augmented batch with low entropy and historical samples in memory. **DPE** [47] gradually aligns the prototypes of the two modalities by optimizing the residual vector to produce more accurate predictions of the streaming test data. **ECALP** [27] constructs a graph over text prompts and test samples and utilizes label propagation to generate improved output. **DynaPrompt** [28] extends the optimization parameters to multiple learnable prompts to simultaneously learn new information and utilize historical knowledge.

We set the batch size to 1 for all online TTA methods in the experiments. Please note that although online methods process the same streaming data, some methods (i.e., DMN, BoostAdapter, DPE, DynaPrompt) employ AugMix augmentation [37] to obtain a batch of views, while other methods simply use single weak augmentation. This distinction should be explicitly recognized to understand the performance differences on some datasets. Since DMN is also able to deploy weak augmentation, we denote this version as $\mathbf{DMN}^W$.

Additionally, while there are numerous recent works [48–54] exploring TTA for VLMs, many of them rely on additional resources, such as large language models [50], generative models [48, 51], or statistical information from ImageNet [49]. To ensure a fair and controlled comparison within our benchmark, we have not included these methods at this stage. We acknowledge the importance of these TTA methods and plan to incorporate them in future versions of TTA-VLM.

## 2.3 Pre-trained Models and Datasets

Following prior work, our benchmark primarily builds on **CLIP** [1]. To assess the generality of TTA methods, we further include **SigLIP** [4], a VLM that retains the dual-encoder architecture but replaces the contrastive softmax loss with a sigmoid-based loss function. This change allows for significantly larger batch sizes during training and yields improved representation learning in some scenarios. To enrich the diversity of base models, we additionally incorporate several training-time

Table 1: Accuracies (%) of TTA methods on **fine-grained datasets** with CLIP-ResNet50.

| | Caltech101 | Pets | Cars | Flowers | Food101 | Aircraft | SUN397 | DTD | EuroSAT | UCF101 | Avg. |
|---|---|---|---|---|---|---|---|---|---|---|---|
| CLIP [1] | 85.88 | 83.62 | 55.75 | 61.67 | 73.96 | 15.69 | 58.82 | 40.43 | 23.68 | 58.90 | 55.84 |
| TPT [21] | 87.91 | 84.68 | 58.39 | 62.08 | **75.03** | 17.16 | **61.31** | 42.43 | **28.41** | 60.64 | **57.80** |
| C-TPT [24] | 87.75 | 83.57 | 56.52 | **64.80** | 74.87 | 16.77 | 60.78 | 41.49 | 26.98 | 60.14 | 57.37 |
| RLCF [39] | **88.15** | 82.77 | 57.87 | 59.16 | 74.30 | 17.16 | 60.56 | 41.31 | 27.23 | **60.93** | 57.07 |
| MTA [40] | 87.30 | **84.82** | 58.59 | 61.02 | 74.28 | **18.06** | 60.74 | 40.31 | 22.53 | 60.59 | 56.82 |
| ZERO [41] | 86.37 | 83.97 | 58.08 | 58.79 | 72.22 | 17.52 | 60.42 | 39.07 | 22.05 | 59.08 | 55.76 |
| TPS [43] | 86.69 | 84.41 | **58.69** | 61.55 | 74.37 | 17.16 | 60.36 | 40.43 | 24.30 | 60.61 | 56.86 |
| R-TPT [25] | 86.33 | 84.08 | 57.87 | 61.35 | 73.44 | 17.61 | 60.58 | 41.55 | 21.40 | 59.50 | 56.37 |
| TDA [26] | 88.15 | 84.19 | 56.81 | 65.16 | 75.22 | 16.53 | 61.19 | 40.78 | 31.15 | 61.80 | 58.10 |
| DMN$^W$ [44] | 85.23 | 84.52 | 56.62 | 64.11 | 74.62 | 16.02 | 59.58 | 41.02 | **37.75** | 60.43 | 57.99 |
| DMN [44] | 86.29 | **85.88** | **59.56** | 61.51 | 74.55 | 18.33 | 61.21 | 41.31 | 31.06 | 61.25 | 58.10 |
| OnZeta [45] | 86.00 | 84.98 | 57.16 | 60.94 | 76.27 | 15.66 | 61.66 | 41.19 | 30.53 | 61.33 | 57.57 |
| BoostAdapter [46] | 87.42 | 84.22 | 58.43 | 65.08 | 74.90 | **18.51** | 61.77 | 41.02 | 32.51 | 62.15 | 58.60 |
| DPE [47] | 88.07 | 84.33 | 58.30 | 63.82 | 74.90 | 15.18 | 60.90 | 42.43 | 25.85 | 62.60 | 57.64 |
| ECALP [27] | **88.48** | 85.50 | 59.05 | **66.18** | **76.31** | 16.92 | **62.40** | **44.27** | 30.22 | **64.50** | **59.38** |
| DynaPrompt [28] | 87.79 | 84.30 | 57.08 | 62.81 | 75.14 | 16.02 | 60.66 | 40.66 | 22.90 | 59.82 | 56.72 |

Table 2: Accuracies (%) of TTA methods on **ImageNet-X datasets** with CLIP-ResNet50.

| | ImageNet | ImageNet-V2 | ImageNet-R | ImageNet-A | ImageNet-Sketch | Avg. | OOD Avg. |
|---|---|---|---|---|---|---|---|
| CLIP [1] | 58.15 | 51.52 | 56.09 | 21.84 | 33.34 | 44.19 | 40.70 |
| TPT [21] | 60.74 | **54.85** | **58.97** | 26.45 | 35.05 | 47.21 | 43.83 |
| C-TPT [24] | 60.38 | 54.27 | 57.76 | 24.07 | 34.73 | 46.24 | 42.71 |
| RLCF [39] | 60.22 | 54.28 | 58.48 | 28.94 | 34.97 | 47.38 | 44.17 |
| MTA [40] | 60.39 | 54.20 | 58.40 | 27.76 | **35.18** | 47.19 | 43.89 |
| ZERO [41] | 60.36 | 54.50 | 57.84 | **29.87** | 34.76 | **47.47** | **44.24** |
| TPS [43] | 59.96 | 53.82 | 58.34 | 28.11 | 34.92 | 47.03 | 43.80 |
| R-TPT [25] | **60.81** | 54.64 | 57.71 | 27.95 | 34.01 | 47.02 | 43.58 |
| TDA [26] | 59.94 | 52.43 | 57.48 | 22.88 | 36.18 | 45.78 | 42.24 |
| DMN$^W$ [44] | 58.65 | 51.47 | 55.99 | 21.88 | 34.74 | 44.55 | 41.02 |
| DMN [44] | 60.52 | **53.85** | 57.32 | **30.55** | 36.19 | 47.69 | 44.48 |
| BoostAdapter [46] | **60.92** | 53.74 | **59.28** | 29.20 | 37.21 | **48.07** | **44.86** |
| ECALP [27] | 60.50 | 52.39 | 57.87 | 22.29 | **37.26** | 46.06 | 42.45 |
| DynaPrompt [28] | 60.15 | 53.31 | 58.24 | 24.85 | 34.53 | 46.22 | 42.73 |

fine-tuning approaches, including **CoOp** [6] and **MaPLe** [32]. We also incorporate **TeCoA** [33], an adversarial fine-tune method designed to enhance model robustness.

As for datasets, we adopt 10 fine-grained classification datasets (**Caltech101** [55], **Pets** [56], **Cars** [57], **Flowers** [58], **Food101** [59], **Aircraft** [60], **SUN397** [61], **DTD** [62], **EuroSAT** [63], **UCF101** [64]) and five imagenet-related datasets (**ImageNet** [65], **ImageNet-V2** [66], **ImageNet-R** [67], **ImageNet-A** [68], **ImageNet-Sketch** [69]). Comprehensive details of each dataset can be found in the supplementary material.

## 3 Vanilla Test-Time Adaptation under Fair Comparison

**Experiment setup.** To ensure a fair and consistent evaluation of diverse TTA methods, we establish a unified benchmarking framework. All components of the experimental pipeline—except the core algorithmic logic specific to each method—are standardized across methods. (**Data**). In the episodic adaptation setting [21], we use a fixed augmentation protocol for each test sample. Specifically, each test batch consists of 64 views of the test data: the first view is a weakly augmented version used for prediction by certain methods, while the remaining 63 views are generated using AugMix [37]. In the online adaptation setting [26], we fix the order of the test stream and clearly state the different data augmentation protocols for all methods. (**Model**). We evaluate all methods using two VLMs, CLIP [1] and SigLIP [4]. Using consistent initialization across all methods allows us to clearly figure out the contribution of the adaptation procedure itself. Additionally, the introduction of SigLIP helps us to assess the generalizability of TTA methods beyond the commonly used CLIP framework. (**Text templates**). To eliminate variability due to prompt selection, we fix the text template to CLIP's default prompt, "a photo of a [CLASS]" across all experiments. For methods capable of leveraging multiple templates, we also evaluate their performance under a multi-template setting to explore potential improvements. (**Hyperparameters**). We adopt the hyperparameters and optimization protocols recommended by the original implementations of each TTA method. However, we note that hyperparameter tuning during test time remains an open and challenging problem.

Table 3: Accuracies (%) of episodic TTA methods on all datasets with **SigLIP (ViT-B/16)**.

| | Fine-grained Avg. | ImageNet | ImageNet-V2 | ImageNet-R | ImageNet-A | ImageNet-Sketch | Avg. | OOD Avg. |
|---|---|---|---|---|---|---|---|---|
| SigLIP [4] | 73.89 | 75.69 | 68.43 | 89.30 | 45.05 | 66.57 | 69.01 | 67.34 |
| TPT [21] | **74.12** | 76.43 | 69.25 | 89.86 | 46.73 | 67.09 | 69.87 | 68.23 |
| C-TPT [24] | 73.78 | 76.11 | 68.97 | 89.57 | 45.76 | 66.94 | 69.47 | 67.81 |
| RLCF [39] | 68.94 | 70.81 | 63.52 | 85.26 | 42.23 | 60.65 | 64.49 | 62.92 |
| MTA [40] | 73.43 | 77.72 | 70.81 | 90.94 | 57.96 | 67.51 | 72.99 | 71.81 |
| ZERO [41] | 71.41 | 77.60 | **70.97** | 90.48 | **61.81** | 66.48 | **73.47** | **72.44** |
| TPS [43] | 73.68 | **78.05** | 70.92 | **91.50** | 58.53 | **67.73** | 73.35 | 72.17 |
| R-TPT [25] | 71.61 | 76.95 | 69.79 | 89.16 | 54.00 | 65.53 | 71.09 | 69.62 |

**Results on CLIP.** We present the adaptation results on CLIP-ResNet50 in Tables 1, 2. Additional results for CLIP-ViT-B/16 and CLIP-ViT-B/32 are included in the supplementary material. Notably, an early work, TPT achieves the highest accuracy of 55.7% on the fine-grained datasets with ResNet50 and remains competitive on ViT-B/16. This indicates that recent methods have marginal accuracy improvement under fair and consistent evaluation settings. However, TPT performs less competitively on ViT-B/32, particularly on ImageNet-related benchmarks. This discrepancy highlights the importance of evaluating TTA methods across diverse model architectures to assess their generalizability. Another interesting observation is that online TTA methods consistently outperform episodic ones on fine-grained tasks, suggesting that leveraging historical context is particularly beneficial in such datasets. Conversely, the introduction of AugMix augmentation allows episodic methods and some online methods to achieve attractive results in ImageNet-related tasks. This implies that different tasks have different preferred strategies.

**Results on SigLIP.** To assess the generality of TTA approaches beyond CLIP, we evaluate episodic TTA methods on SigLIP, with results shown in Table 3. It can be seen that the performance gains on SigLIP are limited. While certain methods yield a 3–4% improvement on ImageNet-related datasets, most TTA methods fail to surpass the zero-shot baseline on fine-grained tasks. These results underscore the need for broader evaluation across VLM architectures and emphasize the lack of generality of current TTA methods when applied to models pre-trained with different objectives.

**Improvement by multiple templates.** We further analyze the impact of using multiple personalized templates and provide the results on the DTD dataset [62] in Figure 2. Most methods benefit from the inclusion of multiple templates. For example, ZERO achieves a 3.1% gain, which is higher than the 0.94% gain of CLIP, highlighting the potential of template diversity.

There are also some online methods that show a drop in performance after introducing multiple templates, even though these templates have been verified to be beneficial for zero-shot classification. While our benchmark mainly focuses on single-template evaluation, this result indicates that the ability to combine with multiple templates is really valuable for TTA methods. These methods have the potential to produce significant improvements when combined with templates generated by large language models tailored to specific tasks or domains.

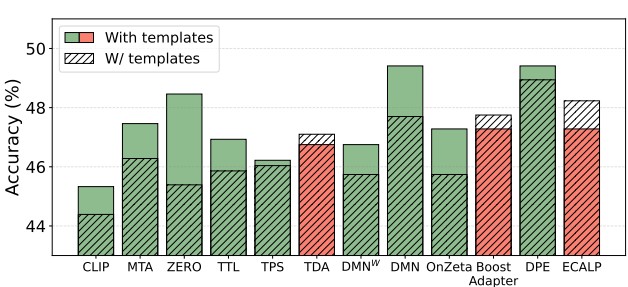

Figure 2: Results (%) of adaptation methods combined with multiple templates on DTD dataset with CLIP-ViT-B/16.

> **Takeaways**
>
> 1. When evaluated under standardized and fair conditions, existing episodic TTA methods provide only limited improvements over early test-time prompting (TPT) baselines.
> 2. Most current TTA methods exhibit limited generalizability and weak compatibility with VLMs beyond CLIP, such as SigLIP.
> 3. Using diverse text templates enhances CLIP's performance and benefits to TTA methods.

Table 4: Accuracies (%) of TTA methods on **fine-grained datasets** with **CoOp** (ResNet50).

| | Caltech101 | Pets | Cars | Flowers | Food101 | Aircraft | SUN397 | DTD | EuroSAT | UCF101 | Avg. |
|---|---|---|---|---|---|---|---|---|---|---|---|
| CoOp [1] | 86.37 | 86.94 | 55.50 | 61.63 | 75.62 | 15.09 | 58.19 | 37.29 | 26.37 | 59.03 | 56.20 |
| TPT [21] | 87.10 | 87.60 | 57.75 | 59.07 | **76.16** | 15.90 | 59.87 | 38.65 | 28.37 | 60.72 | 57.12 |
| C-TPT [24] | 87.18 | 87.00 | 56.81 | **62.00** | 75.67 | 15.48 | 59.12 | 38.30 | 26.86 | 60.14 | 56.86 |
| RLCF [39] | **87.83** | 86.94 | **58.97** | 60.86 | 75.71 | **17.88** | **61.09** | **41.61** | 30.40 | 62.07 | **58.34** |
| MTA [40] | 86.73 | **87.95** | 57.77 | 60.17 | 76.13 | 16.62 | 59.34 | 37.23 | 26.47 | 60.03 | 56.84 |
| ZERO [41] | 86.04 | 87.44 | 56.83 | 57.78 | 74.20 | 16.68 | 58.52 | 36.17 | 26.59 | 59.95 | 56.02 |
| TPS [43] | 86.45 | 87.79 | 57.23 | 59.85 | **76.16** | 15.96 | 59.17 | 38.24 | 27.14 | 60.77 | 56.88 |
| R-TPT [25] | 86.33 | 87.63 | 57.39 | 57.53 | 74.89 | 15.84 | 59.24 | 37.65 | 27.05 | 59.90 | 56.35 |
| TDA [26] | 88.07 | 87.90 | 56.85 | 63.50 | 75.82 | 15.75 | 59.37 | 38.42 | 37.90 | 60.75 | 58.43 |
| DMN$^W$ [44] | 86.17 | 87.79 | 56.39 | 63.38 | 75.86 | 15.57 | 58.98 | 37.88 | 34.64 | 60.11 | 57.68 |
| DMN [44] | 85.84 | **88.50** | 58.55 | 60.09 | 75.84 | **17.07** | 59.34 | 37.53 | 30.00 | 60.80 | 57.36 |
| OnZeta [45] | 86.41 | 86.51 | 57.51 | 60.98 | 76.64 | 14.88 | 59.99 | 37.41 | 30.91 | 60.64 | 57.19 |
| BoostAdapter [46] | **88.11** | 87.90 | 57.78 | 63.22 | 76.25 | 16.77 | 59.81 | 38.48 | **39.17** | 61.06 | **58.86** |
| DPE [47] | 85.27 | 61.43 | 45.80 | 59.16 | 75.69 | 14.25 | 56.04 | 37.00 | 31.49 | 44.01 | 51.01 |
| ECALP [27] | **88.11** | 87.65 | **58.55** | **64.31** | **76.82** | 14.91 | 60.68 | 39.95 | 30.78 | **63.02** | 58.48 |
| DynaPrompt [28] | 87.26 | 84.36 | 57.27 | 62.12 | 74.99 | 16.17 | **60.70** | **41.08** | 22.77 | 60.61 | 56.73 |

Table 5: Accuracies (%) of TTA methods on **fine-grained datasets** with **MaPLe** (ViT-B/16).

| | Caltech101 | Pets | Cars | Flowers | Food101 | Aircraft | SUN397 | DTD | EuroSAT | UCF101 | Avg. |
|---|---|---|---|---|---|---|---|---|---|---|---|
| MaPLe [32] | 91.28 | 89.34 | 64.56 | 66.50 | 83.90 | 22.32 | 64.02 | 43.79 | **50.38** | 70.21 | 64.63 |
| TPT [21] | 91.32 | 89.64 | 66.40 | **67.97** | 84.57 | 23.61 | 64.95 | 45.51 | 48.26 | **70.66** | 65.29 |
| C-TPT [24] | **91.85** | 89.64 | 65.33 | 67.68 | 84.38 | 23.46 | 64.75 | 45.27 | 48.25 | 70.39 | 65.10 |
| RLCF [39] | 91.64 | 84.11 | 66.47 | 63.50 | 84.31 | 21.72 | 61.18 | **46.04** | 42.90 | 67.62 | 62.95 |
| MTA [40] | 91.52 | **89.89** | **67.57** | 66.71 | **84.68** | 24.30 | **65.24** | 45.45 | 48.67 | 70.29 | **65.43** |
| ZERO [41] | 91.76 | 89.37 | 67.16 | 66.14 | 84.08 | 24.15 | 64.85 | 45.09 | 41.85 | 69.07 | 64.35 |
| TTL [42] | 90.99 | 88.96 | 66.00 | 66.34 | 84.15 | **24.66** | 63.85 | 44.80 | 45.33 | 69.23 | 64.43 |
| TPS [43] | 91.36 | 89.56 | 66.93 | 66.71 | 84.66 | 23.13 | 64.78 | 44.80 | 48.23 | 70.31 | 65.05 |
| R-TPT [25] | 91.36 | 89.32 | 67.13 | 67.44 | 84.17 | 24.48 | 65.16 | 44.92 | 41.17 | 68.99 | 64.41 |
| TDA [26] | 92.66 | 89.45 | 66.55 | 68.25 | 84.30 | 23.16 | 65.19 | 45.27 | 53.00 | 70.79 | 65.86 |
| DMN$^W$ [44] | 91.16 | **90.08** | 65.85 | 68.37 | 84.18 | 22.74 | 65.04 | 45.51 | 56.33 | **71.82** | 66.11 |
| DMN [44] | 91.64 | 89.94 | 68.52 | 69.31 | 84.71 | 24.24 | 66.14 | 46.93 | 48.23 | 71.03 | 66.07 |
| OnZeta [45] | 91.56 | 89.94 | **68.55** | 68.62 | 84.95 | **24.99** | **66.77** | 44.80 | 53.79 | **71.82** | 66.58 |
| BoostAdapter [46] | 92.21 | 89.26 | 67.67 | 68.17 | 84.81 | 24.03 | 65.38 | 45.27 | 52.89 | 71.13 | 66.08 |
| DPE [47] | **93.96** | 89.45 | 67.79 | 70.24 | 84.35 | 20.82 | 65.78 | 45.15 | 52.51 | 71.79 | 66.18 |
| ECALP [27] | 91.93 | 89.86 | 68.21 | **73.37** | **85.55** | 22.95 | 66.25 | 46.69 | **59.95** | **73.12** | **67.79** |

## 4 Collaboration with Training-time Fine-tuning Methods

**Experiment setup.** In this section, we investigate the collaboration between test-time adaptation (TTA) methods and training-time fine-tuning approaches. Training-time methods operate between pre-training and deployment stages, leveraging annotated data to tailor the model to specific tasks or domains. To assess collaboration between them, we evaluate TTA methods on VLMs that have been fine-tuned using three representative training-time approaches: CoOp, MaPLe, and TeCoA. **(CoOp method.)** CoOp [6] introduces learnable textual prompts by combining category names with trainable context vectors and optimizes with the labeled data. We use the publicly available CoOp checkpoint trained with 16-shot ImageNet [65] with a context length of 4. TTA methods are applied without modification, following the same optimization settings as in previous sections. **(MaPLe method.)** MaPLe [32] extends prompt learning to both visual and textual modalities by optimizing token embeddings in early transformer layers of the text encoder and their associated projection layers. We employ the MaPLe checkpoint trained with 16-shot data from ImageNet. To better align TTA methods with MaPLe, we expand the optimization space of prompt-based TTA methods (e.g., TPT) to include all learnable components introduced by MaPLe (i.e., text prompt, image prompt, textual tokens, and linear projectors) to further explore their potential. **(TeCoA method.)** TeCoA [33] enhances the robustness of VLMs through contrastive adversarial training, updating the image encoder without altering the architecture. We use the TeCoA checkpoint trained on ImageNet with an adversarial attack radius of 1.0/255. Since TeCoA does not introduce new components or prompt structures, no additional adjustments are required when deploying TTA methods.

**Results of adaptation on CoOp.** Table 4 reports the performance of various TTA methods applied to the CoOp-tuned VLM on fine-grained classification datasets. Compared to CLIP, TTA methods obtain smaller improvements when applied to CoOp. Among episodic methods, only RLCF achieves a notable gain of 2.14%, while the remaining methods yield improvements below 1%. Although CoOp

Table 6: Accuracies (%) of TTA methods on **fine-grained datasets** with **TeCoA** (ResNet50).

| | Caltech101 | Pets | Cars | Flowers | Food101 | Aircraft | SUN397 | DTD | EuroSAT | UCF101 | Avg. |
|---|---|---|---|---|---|---|---|---|---|---|---|
| TeCoA [33] | 78.30 | 76.04 | 22.42 | **33.58** | 28.00 | 5.82 | **37.09** | 26.18 | **16.56** | 38.30 | 36.23 |
| TPT [21] | 79.59 | 76.37 | **22.52** | 31.47 | 27.17 | 6.45 | 36.96 | 26.48 | 13.60 | 38.96 | 35.96 |
| C-TPT [24] | 79.63 | 75.63 | 22.11 | 33.46 | **28.33** | 6.21 | 36.95 | **27.42** | 16.20 | **39.23** | **36.52** |
| MTA [40] | **79.76** | 76.75 | 21.23 | 31.67 | 26.52 | 6.27 | 36.51 | 25.89 | 14.57 | 37.85 | 35.70 |
| ZERO [41] | 77.93 | 76.59 | 19.76 | 30.09 | 24.73 | **6.78** | 34.71 | 26.12 | 12.25 | 36.29 | 34.53 |
| TPS [43] | 79.63 | **76.91** | 21.70 | 32.48 | 27.28 | 6.00 | 36.87 | 26.18 | 12.85 | 38.36 | 35.83 |
| R-TPT [25] | 76.75 | 75.28 | 19.91 | 29.07 | 24.29 | 6.51 | 34.19 | 25.71 | 12.17 | 36.24 | 34.01 |
| TDA [26] | **82.19** | 77.16 | 29.13 | 38.37 | 33.93 | 6.39 | 41.90 | 29.31 | 17.37 | 43.72 | 39.95 |
| DMN$^W$ [44] | 78.01 | 77.05 | 24.50 | 35.61 | 31.36 | 6.15 | 39.60 | 29.08 | 16.63 | 42.11 | 38.01 |
| DMN [44] | 79.19 | 77.32 | 23.18 | 33.21 | 30.33 | **7.47** | 38.35 | 29.67 | 14.74 | 40.71 | 37.42 |
| OnZeta [45] | 79.19 | 77.32 | 23.18 | 33.21 | 30.33 | **7.47** | 38.35 | 29.67 | 14.74 | 40.71 | 37.42 |
| BoostAdapter [46] | 81.58 | 77.08 | 26.49 | **38.98** | 30.86 | 6.90 | 41.18 | 30.38 | 17.54 | 44.54 | 39.55 |
| DPE [47] | 24.10 | 2.75 | 0.41 | 35.00 | 34.94 | 6.48 | 34.46 | 25.71 | 17.09 | 0.85 | 18.18 |
| ECALP [27] | 81.18 | **77.76** | **29.87** | 37.64 | **34.95** | 7.29 | **42.76** | **31.09** | **23.00** | **46.31** | **41.19** |
| DynaPrompt [28] | 79.88 | 76.10 | 22.21 | 32.64 | 28.07 | 6.06 | 37.40 | 25.65 | 14.80 | 39.02 | 36.18 |

outperforms CLIP slightly in base classification accuracy on these datasets, there is no significant difference between performance after adaptation. This suggests that CoOp, which already corrects decision boundaries during training, leaves limited room for further gains through test-time adaptation.

**Results of adaptation on MaPLe.** The adaptation performance for MaPLe is shown in Table 5. As with CoOp, although the baseline accuracy of MaPLe itself is 0.92% higher than CLIP, both episodic and online TTA methods result in marginal. Four episodic TTA methods (i.e., RLCF, ZERO, TTL, R-TPT) even cause performance degradation. This unexpected trend highlights a potential incompatibility between training-time tuning and test-time adaptation, suggesting that their mechanisms may not align in a complementary manner.

**Results of adaptation on TeCoA.** We provide adaptation results on TeCoA in Table 6. Since RLCF uses the original CLIP as the reward model, its results have little reference value and are omitted. Surprisingly, only CTPT achieves a modest improvement of 0.29%, and all other episodic TTA methods lead to performance degradation when applied to TeCoA. In contrast, most online TTA methods show positive gains over the TeCoA baseline, with ECALP achieving the highest improvement of 4.96%. These results suggest that adapting robust models like TeCoA using single-sample techniques remains a difficult and open challenge.

**Remark.** Regarding whether the deployer needs to introduce a training time phase before the test-time method, we have the following suggestions: When the labeled training data and test data share the same or highly overlapping categories, training-time methods can lead to significant gains. For example, fine-tuning on labeled ImageNet data yields noticeable improvements on downstream tasks such as ImageNet-V2 (4-shot ImageNet tuning w/o TTA: 51.5%->55.6% on ImageNet-V2). Also, when robustness or other specific properties are required, a supervised train-time stage (e.g., adversarial fine-tuning) can provide a more robust initialization for TTA. Conversely, if the labeled data is not strongly correlated with the test distribution, supervised fine-tuning tends to generalize poorly (4-shot ImageNet tuning w/o TTA: 55.8%->56.2% on fine-grained dataset). In such scenarios, we recommend applying TTA directly, as it can better adapt to downstream tasks.

> **Takeaways**
> 1. Both episodic and online TTA methods exhibit poor collaboration with training-time tuned models, in contrast to their stronger performance on vanilla CLIP.
> 2. Almost all episodic TTA methods lead to negative transfer on robust TeCoA models.

## 5   Trustworthiness and Stability from Accuracy-oriented Adaptation

**Experiment setup.** While previous sections focus on classification accuracy, real-world deployment requires models that are also reliable and robust. To this end, we extend our evaluation to cover three key dimensions of trustworthiness: calibration, out-of-distribution (OOD) detection, and adversarial robustness. **(Calibration).** Model calibration assesses the alignment between predicted confidence scores and actual correctness. Overconfident incorrect predictions can mislead downstream systems

Table 7: Expected calibration error (%) of TTA methods on **all datasets** with CLIP-ResNet50.

| | Fine-grained Avg. ↓ | IN ↓ | IN-V2 ↓ | IN-R ↓ | IN-A ↓ | IN-Sketch ↓ | Avg. ↓ |
|---|---|---|---|---|---|---|---|
| CLIP [1] | 5.70 (− 0.00) | 1.97 | 2.98 | 0.98 | 21.28 | 3.13 | 6.07 (− 0.00) |
| TPT [21] | 11.30 (↑ 5.60) | 11.34 | 13.89 | 10.56 | 31.12 | 15.29 | 16.44 (↑ 10.37) |
| C-TPT [24] | 6.61 (↑ 0.91) | 6.81 | 9.48 | 5.52 | 26.79 | 11.44 | 12.01 (↑ 5.94) |
| RLCF [39] | 20.26 (↑ 14.56) | 21.43 | 24.72 | 23.33 | 44.47 | 30.09 | 28.81 (↑ 22.74) |
| MTA [40] | 12.20 (↑ 6.50) | 14.45 | 17.79 | 14.88 | 42.68 | 14.98 | 20.95 (↑ 14.88) |
| TPS [43] | 21.16 (↑ 15.46) | 28.60 | 32.95 | 29.28 | 56.69 | 34.96 | 36.50 (↑ 30.43) |
| R-TPT [25] | 11.32 (↑ 5.62) | 11.45 | 13.47 | 10.01 | 30.09 | 12.88 | 15.58 (↑ 9.51) |
| TDA [26] | 9.21 (↑ 3.51) | 6.25 | 10.69 | 0.88 | 29.07 | 10.74 | 11.53 (↑ 5.46) |
| DMN$^W$ [44] | 16.53 (↑ 10.83) | 16.75 | 25.23 | 18.25 | 46.76 | 22.5 | 25.90 (↑ 19.83) |
| DMN [44] | 12.89 (↑ 7.19) | 13.01 | 20.52 | 14.54 | 33.78 | 18.92 | 20.15 (↑ 14.08) |
| BoostAdapter [46] | 17.25 (↑ 11.55) | 17.97 | 16.93 | 12.82 | 52.19 | 25.09 | 25.00 (↑ 18.93) |
| ECALP [27] | 32.21 (↑ 26.51) | 39.27 | 47.47 | 39.87 | 75.79 | 62.35 | 52.95 (↑ 46.88) |
| DynaPrompt [28] | 7.42 (↑ 1.72) | 6.29 | 9.56 | 6.80 | 28.00 | 10.68 | 12.27 (↑ 6.20) |

Table 8: AUC score (%) of TTA methods on **fine-grained datasets** with CLIP-ResNet50.

| | Caltech101 | Pets | Cars | Flowers | Food101 | Aircraft | SUN397 | DTD | EuroSAT | UCF101 | Avg. ↑ |
|---|---|---|---|---|---|---|---|---|---|---|---|
| CLIP [1] | 87.20 | 68.90 | 56.67 | 73.79 | 80.27 | 31.04 | 70.56 | 64.61 | 56.79 | 72.22 | 66.20 (− 0.00) |
| TPT [21] | 85.11 | 65.89 | 55.19 | 72.90 | 79.07 | 29.38 | 69.15 | 63.57 | 56.25 | 69.46 | 64.60 (↓ 1.61) |
| C-TPT [24] | 88.30 | 65.14 | 55.07 | 73.56 | 80.19 | 30.56 | 69.57 | 62.35 | 59.15 | 71.86 | 65.57 (↓ 0.63) |
| RLCF [39] | 80.20 | 63.71 | 52.99 | 69.78 | 74.93 | 32.76 | 63.50 | 62.05 | 57.99 | 62.89 | 62.08 (↓ 4.12) |
| MTA [40] | 86.71 | 63.91 | 48.32 | 73.50 | 79.14 | 28.87 | 67.09 | 63.67 | 55.23 | 68.99 | 63.54 (↓ 2.66) |
| ZERO [41] | 83.34 | 64.12 | 58.50 | 69.64 | 74.48 | 44.43 | 64.96 | 61.02 | 50.13 | 65.58 | 63.62 (↓ 2.59) |
| TPS [43] | 86.75 | 69.50 | 55.64 | 72.45 | 78.99 | 29.51 | 68.29 | 63.75 | 56.32 | 69.83 | 65.10 (↓ 1.10) |
| R-TPT [25] | 83.00 | 64.77 | 54.49 | 72.57 | 78.01 | 29.33 | 68.76 | 62.23 | 55.07 | 67.77 | 63.60 (↓ 2.61) |

or users. We measure calibration using the Expected Calibration Error (ECE) with 20 equal-width bins. We exclude ZERO from episodic methods in this analysis, as it only provides hard predictions without confidence estimates. **(OOD Detection).** OOD detection evaluates a model's ability to recognize inputs from unknown categories. A reliable model should assign low confidence to OOD samples and high confidence to the samples that belong to candidate categories. We assess this via the Area Under the ROC Curve (AUC), a threshold-independent metric. In the experiment, we discard 50% of the original categories and regard samples that belong to them as OOD samples. For online TTA methods, we focus on how the presence of OOD data affects performance on in-distribution examples, rather than OOD detection performance. **(Adversarial robustness).** Adversarial robustness reflects a model's resilience to inputs perturbed by imperceptible but malicious changes. Due to non-differentiable components in many TTA methods, we use CLIP to generate adversarial examples. We then evaluate the robustness of TTA methods by measuring their defense performance against these transferred adversarial examples. We employ PGD attack [35], using a perturbation radius of $1/255$ with 7 steps for CLIP-ResNet, and $4/255$ with 100 steps for CLIP-ViT.

**Calibration Degradation Induced by TTA.** We report the calibration performance of TTA methods in Table 7. Our results indicate that both episodic and online adaptation techniques generally degrade the calibration of the original CLIP model. Among them, C-TPT demonstrates relatively strong calibration preservation, increasing the calibration error by only 0.91% on fine-grained datasets, owing to its design that explicitly minimizes text feature dispersion. Similarly, R-TPT, DMN, and DynaPrompt introduce only modest calibration deterioration. We attribute the calibration degradation to the overconfidence introduced by TTA methods. In pursuit of higher accuracy, they assign overly high probabilities to predictions, ultimately compromising the model's reliability under uncertainty.

**Reduced OOD Detection Performance under Episodic TTA.** Table 8 presents the out-of-distribution (OOD) detection results for episodic TTA methods. In-distribution accuracy results are included in the supplementary material, and their trends remain largely consistent with the experiments in the early sections. It is shown that there is a 1–4% reduction in AUC scores, which indicates weakened OOD sensitivity. The phenomenon of decreased OOD detection capability is similar to that of calibration. This degradation suggests that TTA methods often assign high confidence to uncertain samples, sometimes even exceeding the in-distribution samples.

**Limited Adversarial Robustness in Episodic TTA.** We provide the adversarial robustness of episodic TTA methods in Table 9. Among them, we find that methods incorporating ensemble strategies outperform those relying solely on the current test sample. This aligns with recent findings

Table 9: Robustness (%) of episodic TTA methods on **fine-grained datasets** with CLIP-ViT-B/16.

| | ImageNet | Caltech101 | Pets | Cars | Flowers | Food101 | Aircraft | SUN397 | DTD | EuroSAT | UCF101 | Avg. |
|---|---|---|---|---|---|---|---|---|---|---|---|---|
| CLIP [1] | 0.00 | 0.16 | 0.00 | 0.00 | 0.00 | 0.00 | 0.00 | 0.00 | 0.00 | 0.00 | 0.00 | 0.02 |
| TPT [21] | 0.00 | 0.24 | 0.00 | 0.00 | 0.00 | 0.00 | 0.00 | 0.00 | 0.06 | 0.00 | 0.00 | 0.03 |
| C-TPT [24] | 0.00 | 0.16 | 0.00 | 0.00 | 0.00 | 0.00 | 0.00 | 0.00 | 0.00 | 0.00 | 0.00 | 0.02 |
| RLCF [39] | 0.00 | 0.28 | 0.00 | 0.00 | 0.00 | 0.00 | 0.00 | 0.00 | 0.30 | 0.00 | 0.00 | 0.06 |
| MTA [40] | 25.15 | 67.30 | 46.31 | 10.25 | 20.34 | 27.46 | 2.04 | 25.28 | 13.59 | 0.32 | 29.50 | 24.24 |
| ZERO [41] | **38.93** | **76.71** | **58.16** | **27.80** | 36.50 | 41.09 | 8.88 | 42.49 | **27.90** | 4.91 | **41.13** | **36.56** |
| TTL [42] | 4.26 | 22.43 | 9.46 | 11.96 | 11.94 | 9.74 | 3.42 | 3.95 | 5.61 | 0.01 | 4.12 | 8.26 |
| TPS [43] | 0.00 | 0.20 | 0.00 | 0.00 | 0.00 | 0.00 | 0.00 | 0.00 | 0.00 | 0.00 | 0.00 | 0.02 |
| R-TPT [25] | 38.61 | 76.11 | 56.75 | 27.57 | **37.27** | **41.52** | **9.00** | **42.65** | 27.90 | **5.30** | 41.13 | 36.52 |

Table 10: Accuracies (%) of online TTA methods on **fine-grained datasets** with CLIP-ResNet50 when adversarial examples are mixed into the test data stream.

| | Caltech101 | | Pets | | Cars | | Flowers | | Food101 | | Aircraft | |
|---|---|---|---|---|---|---|---|---|---|---|---|---|
| | w/ adv. | w adv. | w/ adv. | w adv. | w/ adv. | w adv. | w/ adv. | w adv. | w/ adv. | w adv. | w/ adv. | w adv. |
| CLIP [1] | 86.60 | 86.60 | 84.12 | 84.12 | 56.05 | 56.05 | 62.55 | 62.55 | 73.77 | 73.77 | 15.40 | 15.40 |
| TDA [26] | 88.24 | 86.92 | 84.45 | 82.95 | 56.75 | 56.33 | 65.10 | 65.68 | 74.98 | 74.86 | 15.82 | 16.18 |
| DMN$^W$ [44] | 86.27 | 86.68 | 85.01 | 84.23 | 56.03 | 56.33 | 64.60 | 65.02 | 74.60 | 74.06 | 15.70 | 15.64 |
| DMN [44] | 86.84 | 86.84 | 85.79 | 86.23 | 59.76 | 60.08 | 60.00 | 60.99 | 74.34 | 74.50 | 17.69 | 17.93 |
| OnZeta [45] | 86.92 | 86.18 | 83.79 | 78.68 | 57.77 | 56.43 | 63.29 | 61.73 | 75.89 | 71.67 | 15.58 | 16.24 |
| BoostAdapter [46] | 87.83 | 87.99 | 84.95 | 83.34 | 59.86 | 59.21 | 64.86 | 65.35 | 74.65 | 74.51 | 17.21 | 18.00 |
| DPE [47] | 87.58 | 88.57 | 84.51 | 83.84 | 58.86 | 58.76 | 64.36 | 65.76 | 74.44 | 74.64 | 16.06 | 15.28 |
| ECALP [27] | 88.16 | 87.83 | 85.34 | 83.40 | 66.01 | 55.93 | 66.01 | 65.35 | 76.02 | 74.78 | 16.67 | 17.33 |
| DynaPrompt [28] | 88.40 | 88.49 | 84.40 | 84.73 | 57.54 | 58.49 | 64.53 | 63.29 | 74.94 | 74.93 | 16.85 | 16.24 |

| | SUN397 | | DTD | | EuroSAT | | UCF101 | | Avg. | | |
|---|---|---|---|---|---|---|---|---|---|---|---|
| | w/ adv. | w adv. | w/ adv. | w adv. | w/ adv. | w adv. | w/ adv. | w adv. | w/ adv. | w adv. | Δ |
| CLIP [1] | 58.60 | 58.60 | 40.53 | 40.53 | 23.07 | 23.07 | 58.46 | 58.46 | 55.92 | 55.92 | (− 0.00) |
| TDA [26] | 60.40 | 59.19 | 40.17 | 39.33 | 33.28 | 23.12 | 61.26 | 59.91 | 58.05 | 56.45 | (↓ -1.60) |
| DMN$^W$ [44] | 58.80 | 59.10 | 40.41 | 40.53 | 35.78 | 14.83 | 59.11 | 59.54 | 57.54 | 55.60 | (↓ -1.94) |
| DMN [44] | 59.82 | 60.13 | 42.33 | 42.69 | 32.91 | 27.73 | 59.86 | 60.40 | 57.93 | 57.75 | (↓ -0.18) |
| OnZeta [45] | 60.93 | 60.86 | 40.05 | 38.85 | 31.48 | 28.08 | 60.94 | 59.86 | 57.66 | 55.86 | (↓ -1.81) |
| BoostAdapter [46] | 61.20 | 60.54 | 41.37 | 39.81 | 34.57 | 25.46 | 61.85 | 60.40 | 58.84 | 57.46 | (↓ -1.37) |
| DPE [47] | 60.43 | 59.73 | 42.93 | 39.81 | 24.70 | 27.24 | 61.80 | 59.91 | 57.57 | 57.35 | (↓ -0.21) |
| ECALP [27] | 61.40 | 60.01 | 42.69 | 41.01 | 31.36 | 30.20 | 63.79 | 59.59 | 58.99 | 57.54 | (↓ -1.44) |
| DynaPrompt [28] | 60.51 | 60.73 | 39.93 | 39.33 | 22.26 | 21.66 | 59.38 | 59.81 | 56.77 | 56.87 | (↓ -0.10) |

in literature [25]. However, many methods still fail to defend against adversarial attacks effectively, with several episodic TTA methods exhibiting near-zero robustness on CLIP-ViT.

**Online TTA Stability under OOD and Adversarial Inputs.** We assess the stability of online TTA methods to streaming data perturbed with adversarial examples and report the results in Tables 10. The results under OOD samples can be found in the supplementary material. While the introduction of OOD samples results in minor accuracy degradation for most online methods, OnZeta and DPE suffer from a significant drop of 1.20%, and 2.07%, respectively. In contrast, adversarial examples have a larger effect: 5 out of 8 online methods experience accuracy drops exceeding 1%. We hypothesize that some methods incorrectly save adversarial samples in memory, resulting in inaccurate prototypes.

> **Takeaways**
> 1. Both episodic and online TTA methods cause a decrease in calibration of CLIP.
> 2. Episodic TTA decreases OOD detection ability and exhibits limited adversarial robustness.
> 3. Online TTA methods cause performance degradation when exposed to the data streams mixed with OOD or adversarial samples.

## 6 Conclusion

In this work, we present TTA-VLM, a benchmark designed to evaluate test-time adaptation (TTA) methods for VLMs. Our benchmark addresses key limitations in current TTA research, including inconsistent experimental setups, limited evaluation metrics, and insufficient analysis. By implementing 8 episodic and 7 online TTA methods for VLMs and evaluating them across 15 datasets with CLIP, SigLIP, and training-time tuned models, we provide a fair and comprehensive comparison.

Our experiments reveal several challenges for existing TTA research: (1) the performance gains of most existing TTA methods are marginal when compared to early pioneering baseline TPT, (2) current TTA methods do not collaborate well with training-time fine-tuning methods, and (3) many TTA methods trade off trustworthiness, such as calibration and robustness, for accuracy improvements. These findings highlight the need for more reliable and generalizable TTA strategies that consider broader model qualities beyond classification accuracy.

We hope that TTA-VLM will serve as a useful tool for researchers to better understand TTA for VLMs, promoting more reliable, general, and effective test-time adaptation techniques in the future.

**Limitation.** Our benchmark focuses on classification tasks and does not cover the broader tasks of VLMs, such as image captioning, VQA, and segmentation. Also, our benchmark emphasizes fair comparisons across existing TTA methods, thus, we do not include TTA methods that leverage additional resources such as generative models, LLM, or statistical information of ImageNet.

## Acknowledgments and Disclosure of Funding

Funding in direct support of this work was provided by the National Natural Science Foundation of China under Grants (62276256, U2441251), the Young Elite Scientists Sponsorship Program by CAST (2023QNRC001), and the Young Scientists Fund of the State Key Laboratory of Multimodal Artificial Intelligence Systems (ES2P100117).

The authors declare that they have no competing interests.

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

# A Introduction of TTA Methods

In this section, we introduce the techniques of the methods included in the benchmark in detail. And the official code links for all methods are listed in Table 11.

**TPT** [21] applies the AugMix augmentation strategy to a test sample, generating a batch composed of 63 strongly augmented views alongside one weakly augmented view. Using CLIP, denoted as $f(\cdot)$, TPT computes logits for all views and selects those with the lowest entropy. The textual prompt parameters $\theta_p$ are then optimized by minimizing the marginal entropy over the selected views:

$$\min_{\theta_p} \mathcal{L}_{TPT} = \mathcal{H}(\bar{p}) = \mathcal{H}(\frac{1}{|\mathcal{B}|} \sum_{x \in \mathcal{B}} f(x)), \tag{1}$$

where $\mathcal{B}$ denotes the set of selected views, and $\mathcal{H}$ is the Shannon entropy. After optimization, the final prediction is obtained from the weakly augmented view using the updated prompt.

**C-TPT** [24] is a test-time adaptation method designed to simultaneously enhance prediction accuracy and calibration for vision-language models. Researchers observe that conventional test-time prompt tuning often leads to high dispersion among text features, which correlates with increased calibration errors. Therefore, C-TPT introduces an additional regularization term, Average Text Feature Dispersion, to encourage clustering of class-wise text features, thereby reducing calibration error. The full loss function is defined as:

$$\min_{\theta_p} \mathcal{L} = \mathcal{L}_{TPT} + \mathcal{L}_{CTPT} = \mathcal{L}_{TPT} + \sum_{k=1}^{C} ||\bar{t} - t_k||, \tag{2}$$

where $t_k$ denotes the text feature for the $k$-th class, $\bar{t}$ is the mean of all class text features, and $C$ is the number of candidate classes.

**RLCF** [39] integrates a CLIP model as a reward function to guide adaptation through reinforcement learning. Given a single test sample, the model generates output candidates via sampling, and the CLIP reward model evaluates these candidates based on their alignment with the input. The VLM is then adapted to maximize this feedback signal, effectively encouraging semantically meaningful predictions rather than simply confident ones. Formally, the adaptation objective seeks to optimize the expected reward under the output distribution of the model, using task-specific sampling and a reward baseline to stabilize training.

**MTA** [40] is a training-free TTA method designed to improve the performance without relying on prompt tuning or hand-crafted filtering rules. Unlike traditional test-time prompt tuning methods that require optimization of textual prompt parameters, MTA operates directly on augmented image views using a robust density estimation framework. Since it does not rely on gradient backpropagation, MTA can also be applied in black-box settings. Specifically, MTA introduces an inlierness score for each augmented view, which quantifies its contribution to the final prediction. These scores are jointly optimized with a MeanShift-based mode seeking procedure to identify high-density regions in the visual embedding space.

**ZERO** [41] is a forward-pass-only episodic TTA method designed to address the inefficiency of conventional prompt tuning approaches. Like MTA, ZERO performs adaptation without any gradient updates. ZERO generates $N$ strong augmentations of a test sample, computing predictions for each, and selecting the most confident ones based on maximum softmax probabilities. Then, it performs marginalization over the retained predictions using a zero-temperature softmax, effectively turning the prediction into an argmax operation over logits.

**TTL** [42] is a parameter-efficient TTA method designed to improve VLM's performance. Unlike prompt tuning-based methods, TTL introduces low-rank adapters into the attention layers of the image encoder, while keeping both the prompts and the text encoder frozen. During adaptation, TTL applies strong data augmentations to the input and enforces prediction consistency using a self-supervised weighted entropy loss. The objective encourages confident and consistent predictions across augmented views, formulated as:

$$\min_{\theta_{\text{LoRA}}} \mathcal{L} = \frac{1}{|\mathcal{B}|} \sum_{x \in \mathcal{B}} \beta(x)\mathcal{H}(f(x)) = \frac{1}{|\mathcal{B}|} \sum_{x \in \mathcal{B}} \frac{1}{exp(-\mathcal{H}(f(x)) - \epsilon)} \mathcal{H}(f(x)), \tag{3}$$

where $\epsilon$ is a normalization factor and $\mathcal{B}$ denotes the batch generated by augmentation algorithms.

**TPS** [43] chooses to minimize the marginal entropy through optimizing the shift vector in the embedding space instead of the textual prompt. This process requires no labeled data and incurs minimal computational and memory overhead.

**R-TPT** [25] addresses the vulnerability of existing test-time adaptation methods to adversarial perturbations. Instead of relying on marginal entropy minimization, R-TPT focuses on minimizing point-wise entropy for each selected augmented view, thereby avoiding conflict introduced by the Kullback-Leibler (KL) divergence term. The objective is defined as:

$$\min_{\theta_{\text{prompt}}} \mathcal{L}_{RTPT} = \frac{1}{|\mathcal{B}|} \sum_{x \in \mathcal{B}} \mathcal{H}(f(x)), \tag{4}$$

where $\mathcal{B}$ denotes the subset of test-time augmentations with low-entropy predictions. To further enhance robustness, R-TPT incorporates a reliability-based ensembling strategy, which assigns higher weights to augmented views that are more consistent in their predictions. This leads to final predictions that are both accurate and resistant to adversarial noise.

**TDA** [26] is a training-free and backpropagation-free online test-time adaptation method designed to enable efficient adaptation of vision-language models. Specifically, TDA maintains a dynamic memory queue by saving the high-confidence samples to construct a positive cache and the low-confidence samples to construct a negative cache for each category. As test samples arrive, TDA updates the cache and uses it for progressive pseudo-label refinement, improving label quality without any parameter updates.

**DMN** [44] is a versatile online TTA method designed to support zero-shot, few-shot, and training-free classification tasks. It introduces a dynamic memory, which accumulates test samples' features on the fly to exploit additional information during inference. During inference, the model leverages the memory bank to refine predictions by dynamically retrieving and aggregating relevant features.

**OnZeta** [45] introduces a TTA framework designed for online scenarios, where each test image arrives sequentially and is classified without storing past data. OnZeta proposes two key components: online label learning and online proxy learning. The former dynamically estimates the label distribution of the target data stream to reflect its evolving characteristics. The latter updates the vision-space class proxies to reduce the modality gap between image and text features, thereby improving alignment. Formally, OnZeta combines predictions from both components to produce a final classification output.

**BoostAdapter** [46] is a lightweight TTA method that integrates both instance-agnostic and instance-aware information retrieval. BoostAdapter introduces a memory-based strategy that balances efficiency and adaptivity. Specifically, it maintains a key-value memory that stores features from two types of samples: historical samples, which are selected from the test data stream to capture general distributional patterns of the target domain, and boosting samples, which are generated through regional bootstrapping to reflect the characteristics of the current test instance. The final prediction is produced by combining efficient feature retrieval with test sample-specific enhancement.

**DPE** [47] is an online TTA method that leverages multi-modal information and accumulates task-specific knowledge over time. DPE maintains and updates two sets of class prototypes—textual and visual—to better capture the semantics of target classes. To ensure alignment between modalities, DPE introduces learnable residuals for each test instance, which are jointly optimized to promote consistency between the textual and visual prototypes. These prototypes evolve continuously, enabling the model to refine its understanding of class representations throughout the test phase.

**ECALP** [27] introduces a graph-based method without requiring task-specific tuning or additional labeled data. DPE dynamically constructs a graph comprising test samples, few-shot examples, and class-level text prompts. It performs label propagation over this graph to exploit the underlying test data manifold, enabling label-efficient adaptation. Crucially, it introduces a context-aware feature re-weighting mechanism that adjusts node similarities to improve adaptation performance across tasks.

**DynaPrompt** [28] enhances online test-time adaptation by addressing limitations of traditional test-time prompt tuning. DynaPrompt maintains a dynamic prompt buffer that stores multiple prompts and updates one of them during inference. For each incoming test sample, it selects relevant prompts from the buffer using a dual-criteria strategy based on prediction entropy and probability difference. To further handle unseen distribution shifts, DynaPrompt introduces a dynamic prompt appending mechanism, which allows new prompts to be added to the buffer while removing inactive ones.

Table 11: Overview of methods used in TTA-VLM.

| Method | Venue | Code |
|---|---|---|
| TPT [55] | NeurIPS'22 | https://github.com/azshue/TPT |
| C-TPT [24] | ICLR'24 | https://github.com/hee-suk-yoon/C-TPT |
| RLCF [39] | ICLR'24 | https://github.com/mzhaoshuai/RLCF |
| MTA [40] | CVPR'24 | https://github.com/MaxZanella/MTA |
| ZERO [41] | NeurIPS'24 | https://github.com/FarinaMatteo/zero |
| TTL [42] | WACV'25 | https://github.com/Razaimam45/TTL-Test-Time-Low-Rank-Adaptation |
| TPS [43] | WACV'25 | https://github.com/elaine-sui/TPS |
| R-TPT [25] | CVPR'25 | https://github.com/TomSheng21/R-TPT |
| TDA [26] | CVPR'24 | https://github.com/kdiAAA/TDA |
| DMN [44] | CVPR'24 | https://github.com/YBZh/DMN |
| OnZeta [45] | ECCV'24 | https://github.com/idstcv/OnZeta |
| BoostAdapter [46] | NeurIPS'24 | https://github.com/taolinzhang/BoostAdapter |
| DPE [47] | NeurIPS'24 | https://github.com/zhangce01/DPE-CLIP |
| ECALP [27] | ICLR'25 | https://github.com/Yushu-Li/ECALP |
| DynaPrompt [28] | ICLR'25 | https://github.com/zzzx1224/DynaPrompt |

Table 12: Overview of datasets used in TTA-VLM.

| Dataset | Description | # Classes | # Test |
|---|---|---|---|
| Caltech101 [55] | Classic object recognition dataset with 101 categories, with varied poses and backgrounds | 100 | 2,465 |
| Pets [56] | Images of 37 cat and dog breeds with high variability in pose and lighting | 37 | 3,669 |
| Cars [57] | Fine-grained car models from different manufacturers | 196 | 8,041 |
| Flowers [58] | Images of 102 flower species, captured in diverse lighting and angles | 102 | 2,463 |
| Food101 [59] | Real-world images of 101 popular food categories, with high visual diversity | 101 | 30,300 |
| Aircraft [60] | Aircraft variants with fine-grained differences in shape and appearance | 100 | 3,333 |
| SUN397 [61] | Scene classification dataset with a wide range of indoor and outdoor scenes | 397 | 19,850 |
| DTD [62] | Textures described by various attributes (e.g., "striped", "bumpy") | 47 | 1,692 |
| EuroSAT [63] | Multi-spectral satellite images capturing diverse land use classes | 10 | 8,100 |
| UCF101 [64] | Frames extracted from videos covering 101 human actions | 101 | 3,783 |
| ImageNet [65] | Large-scale dataset with diverse object and scene categories | 1,000 | 50,000 |
| ImageNet-A [68] | Natural adversarial samples misclassified by standard models | 200 | 7,500 |
| ImageNet-V2 [66] | A curated set of new test samples closely matching original ImageNet distribution | 1,000 | 10,000 |
| ImageNet-R [67] | Rendered versions (e.g., cartoons, 3D models, paintings) of ImageNet classes | 200 | 30,000 |
| ImageNet-S [69] | Sketch-style line drawings representing the ImageNet classes | 1,000 | 50,889 |

**TTA methods beyond classification.** Although our benchmark focuses on classification tasks, several recent works explore TTA in broader visual domains such as segmentation, detection, anomaly localization, and video understanding. We briefly introduce their core ideas below.

Among episodic TTA approaches, AnoCLIP [70] proposes a zero-shot anomaly localization framework that enhances CLIP's patch-level alignment and introduces a lightweight test-time adapter for refining localization. T3AL [71] performs episodic test-time adaptation for zero-shot temporal action localization by generating video-level pseudo-labels and refining temporal regions using a self-supervised strategy. DTS-TPT [72] extends episodic test-time prompt tuning to video activity recognition, aligning text prompts with multi-scale temporal features for improved zero-shot accuracy. CLIP-DIY [73] achieves training-free open-vocabulary semantic segmentation by aggregating CLIP's patch-level predictions with unsupervised localization guidance at test time. CLIPtrase [74] recalibrates CLIP's patch correlations to improve local feature discrimination, boosting training-free semantic segmentation performance. VocAda [75] introduces a plug-and-play vocabulary adapter for open-vocabulary detection that refines user-defined labels through image captioning and noun selection at test time. As for online TTA, TTCS [76] adapts the Segment Anything Model for medical image segmentation using CLIP-guided prompt generation and adaptive self-training. TEST-V [77] combines prompt-based and support-set adaptation for zero-shot video classification, tuning support samples dynamically to enhance temporal consistency. These methods show that TTA is expanding beyond classification toward dense and video tasks, highlighting the versatility of adaptation-based paradigms for broader vision-language applications.

## B Introduction of Datasets

We introduce the detailed information of the datasets included in the benchmark in Table 12 and provide representative examples in Figure 3.

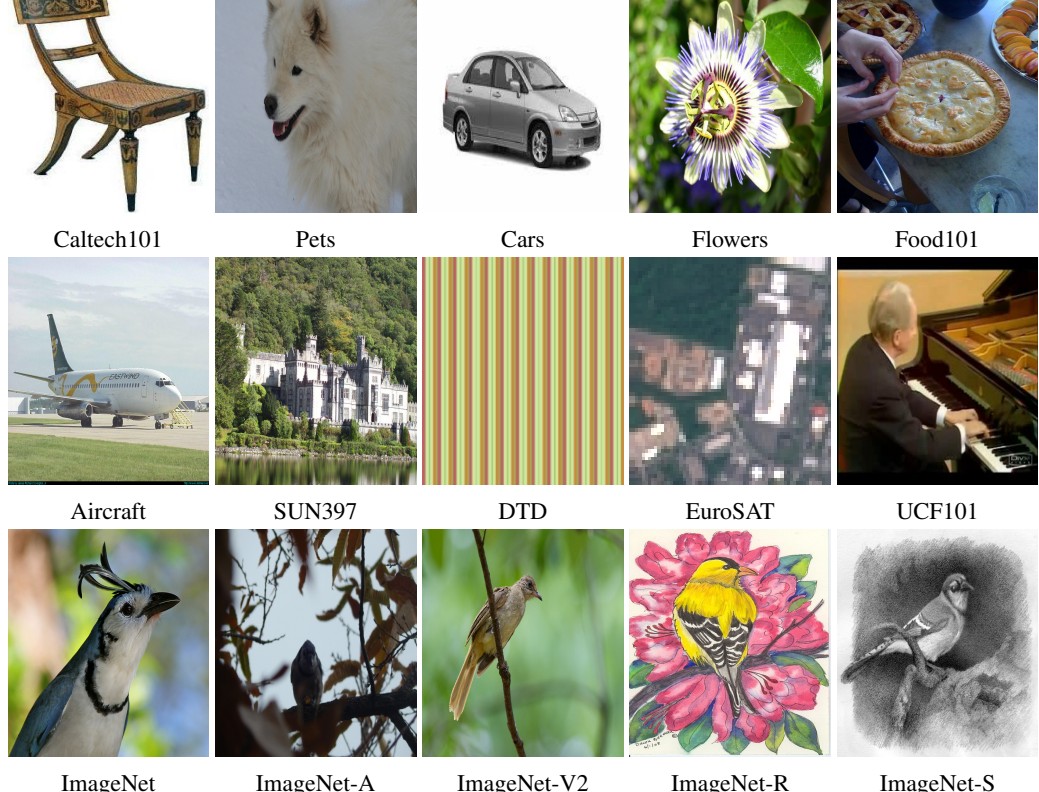

Figure 3: Examples from the 15 datasets used in TTA-VLM, covering objects, animals, scenes, actions, textures, and cross-domain visual tasks.

## C  Additional Experimental Results

**Time and GPU Memory Usage.** While the main text focuses on evaluating the performance of TTA methods across multiple effectiveness metrics, it is equally critical to assess their computational efficiency. Time and resource constraints are crucial considerations for practical deployment, especially on edge devices or limited-resource environments.

To this end, we benchmark the runtime and GPU memory consumption of all TTA methods using 1,000 ImageNet test samples. All evaluations are conducted on an NVIDIA A6000 GPU with 48 GB memory, employing automatic mixed precision (AMP) to reflect optimized inference conditions. Detailed results are reported in Table 13. Our findings show that methods involving optimization of textual prompts generally incur higher computational costs. For instance, TPT [21] requires 441.5 seconds and 14.7 GB of GPU memory to process 1,000 test samples. More resource-intensive methods like DynaPrompt, which simultaneously optimize multiple prompts, consume nearly three times the memory and time. These results indicate that designing lightweight, efficient TTA methods remains a key challenge for enabling real-world deployment.

## D  Detailed Experimental Results

**Accuracy on CLIP.** We report the accuracy of both episodic and online TTA methods evaluated on CLIP-ViT-B/16 and CLIP-ViT-B/32. Results are presented in Tables 14, 15 for ViT-B/16, and Tables 16, 17 for ViT-B/32 across fine-grained and large-scale datasets.

**Accuracy on SigLIP.** To assess the generality of TTA methods, we further evaluate all methods on SigLIP-ViT-B/16. Accuracy results are summarized in Table 18.

Table 13: Time and GPU Memory Usage of TTA methods on 1000 **ImageNet** test samples with **CLIP-ViT-B/16**.

| | Adapt Time (s / 1000 samples) | GPU Memory Usage (GB) |
|---|---|---|
| CLIP [32] | 70.5 | 1.4 |
| TPT [21] | 441.5 | 14.7 |
| C-TPT [24] | 442.2 | 14.7 |
| RLCF [39] | 920.5 | 15.5 |
| MTA [40] | 124.4 | 1.5 |
| ZERO [41] | 384.3 | 1.6 |
| TTL [42] | 184.6 | 6.7 |
| TPS [43] | 62.1 | 1.5 |
| R-TPT [25] | 598.3 | 14.7 |
| TDA [26] | 73.9 | 1.4 |
| $DMN^W$ [44] | 248.9 | 16.4 |
| DMN [44] | 360.9 | 16.5 |
| OnZeta [45] | 72.6 | 31.1 |
| BoostAdapter [46] | 188.2 | 1.7 |
| DPE [47] | 143.1 | 16.4 |
| ECALP [27] | 77.5 | 1.4 |
| DynaPrompt [28] | 1157.1 | 43.0 |

Table 14: Accuracies (%) of TTA methods on **fine-grained datasets** with CLIP-ViT-B/16.

| | Caltech101 | Pets | Cars | Flowers | Food101 | Aircraft | SUN397 | DTD | EuroSAT | UCF101 | Avg. |
|---|---|---|---|---|---|---|---|---|---|---|---|
| CLIP [1] | 94.00 | 88.25 | 65.51 | 67.40 | 83.64 | 23.91 | 62.56 | 44.39 | 42.22 | 65.24 | 63.71 |
| TPT [21] | 94.24 | 87.22 | 66.24 | 68.62 | 84.66 | 23.55 | 65.45 | 47.04 | 43.04 | 68.41 | 64.85 |
| C-TPT [24] | 93.71 | 88.14 | 65.71 | 69.43 | 83.17 | 23.94 | 64.58 | 45.27 | 42.47 | 64.97 | 64.14 |
| RLCF [39] | 94.44 | 86.97 | 66.41 | 68.29 | 84.22 | 22.26 | 65.27 | 46.04 | 43.47 | 66.96 | 64.43 |
| MTA [40] | 94.40 | 87.90 | 67.49 | 67.72 | 84.45 | 24.84 | 65.28 | 46.28 | 42.46 | 67.80 | 64.86 |
| ZERO [41] | 94.00 | 87.33 | 67.03 | 67.44 | 83.78 | 24.81 | 65.59 | 45.39 | 37.28 | 66.53 | 63.92 |
| TTL [42] | 93.75 | 87.22 | 66.25 | 66.38 | 83.99 | 24.75 | 65.07 | 45.86 | 39.02 | 67.30 | 63.96 |
| TPS [43] | 94.16 | 87.44 | 67.21 | 67.64 | 84.40 | 24.78 | 64.68 | 46.04 | 42.56 | 67.46 | 64.64 |
| R-TPT [25] | 93.91 | 86.73 | 66.67 | 69.02 | 84.28 | 24.03 | 65.50 | 46.16 | 34.93 | 67.35 | 63.86 |
| TDA [26] | 94.73 | 88.83 | 66.34 | 69.87 | 84.29 | 24.18 | 65.71 | 47.10 | 53.25 | 68.57 | 66.29 |
| $DMN^W$ [44] | 94.04 | 89.15 | 66.83 | 70.20 | 84.11 | 24.21 | 64.27 | 45.74 | 55.31 | 67.91 | 66.18 |
| DMN [44] | 92.78 | 88.44 | 68.65 | 69.39 | 84.56 | 25.02 | 66.62 | 47.70 | 55.78 | 68.70 | 66.76 |
| OnZeta [45] | 93.96 | 89.18 | 66.43 | 68.66 | 84.72 | 23.79 | 63.97 | 46.16 | 52.86 | 67.75 | 65.75 |
| BoostAdapter [46] | 94.56 | 88.63 | 68.03 | 70.00 | 84.61 | 25.38 | 66.43 | 47.75 | 54.12 | 69.18 | 66.87 |
| DPE [47] | 94.44 | 89.04 | 67.84 | 70.24 | 83.82 | 24.12 | 64.58 | 48.94 | 52.53 | 68.65 | 66.42 |
| ECALP [27] | 93.55 | 89.67 | 68.16 | 72.72 | 85.64 | 25.65 | 68.05 | 48.23 | 55.69 | 72.11 | 67.95 |
| DynaPrompt [28] | 94.24 | 87.84 | 66.89 | 69.02 | 84.67 | 24.21 | 64.85 | 46.10 | 41.89 | 67.91 | 64.76 |

**Calibration on fine-grained datasets.** We provide a detailed evaluation of model calibration for both episodic and online TTA methods using CLIP-ResNet50 as the backbone. Calibration results are reported in Table 19.

**Clean accuracy performance of Episodic TTA under the invasion of OOD samples.** We provide accuracy results of clean samples for episodic TTA methods under the invasion of OOD samples in Table 20.

**Stability of online TTA under OOD samples.** To evaluate the stability of online adaptation, we assess the performance of online TTA methods when exposed to streams containing out-of-distribution (OOD) samples. The corresponding results are shown in Table 21.

**Visualization of visual features.** We employ t-SNE to visualize the visual representations learned by CLIP (ViT-B/32) [1] and TTL [42] on a subset of the UCF101 dataset. As shown in Figure 4, the features generated by TTL exhibit more compact and well-separated clusters compared to the original zero-shot features from CLIP, with fewer ambiguous samples lying near decision boundaries. This indicates that TTL produces more discriminative and robust representations, consistent with its superior classification accuracy and adversarial robustness observed in quantitative experiments.

Table 15: Accuracies (%) of TTA methods on **ImageNet-X datasets** with CLIP-ViT-B/16.

| | ImageNet | ImageNet-V2 | ImageNet-R | ImageNet-A | ImageNet-Sketch | Avg. | OOD Avg. |
|---|---|---|---|---|---|---|---|
| CLIP [1] | 66.72 | 60.84 | 73.99 | 47.80 | 46.15 | 59.10 | 57.20 |
| TPT [21] | 68.94 | 63.38 | 77.13 | 54.75 | 47.92 | 62.42 | 60.80 |
| C-TPT [24] | 68.52 | 62.60 | 75.87 | 51.35 | 47.48 | 61.16 | 59.33 |
| RLCF [39] | 68.56 | 63.02 | 77.08 | 57.39 | 47.98 | 62.81 | 61.37 |
| MTA [40] | 69.24 | 63.60 | 77.00 | 57.03 | 48.48 | 63.07 | 61.53 |
| ZERO [41] | 69.28 | 64.16 | 77.38 | 59.77 | 48.59 | 63.84 | 62.48 |
| TTL [42] | 69.28 | 64.36 | 77.76 | 59.00 | 48.75 | 63.83 | 62.47 |
| TPS [43] | 68.83 | 63.68 | 76.98 | 58.19 | 48.24 | 63.18 | 61.77 |
| R-TPT [25] | 69.37 | 63.98 | 76.93 | 57.72 | 47.75 | 63.15 | 61.60 |
| TDA [26] | 68.28 | 61.24 | 75.36 | 49.16 | 48.73 | 60.55 | 58.62 |
| DMN$^W$ [44] | 67.61 | 60.86 | 74.22 | 48.05 | 47.98 | 59.74 | 57.78 |
| DMN [44] | 69.67 | 63.81 | 76.82 | 59.48 | 49.97 | 63.95 | 62.52 |
| BoostAdapter [46] | 69.33 | 63.09 | 77.30 | 58.15 | 49.60 | 63.49 | 62.04 |
| ECALP [27] | 69.40 | 61.40 | 75.81 | 47.36 | 50.87 | 60.97 | 58.86 |
| DynaPrompt [28] | 68.76 | 62.72 | 76.67 | 53.01 | 47.58 | 61.75 | 60.00 |

Table 16: Accuracies (%) of TTA methods on **fine-grained datasets** with CLIP-ViT-B/32.

| | Caltech101 | Pets | Cars | Flowers | Food101 | Aircraft | SUN397 | DTD | EuroSAT | UCF101 | Avg. |
|---|---|---|---|---|---|---|---|---|---|---|---|
| CLIP [1] | 91.36 | 85.06 | 60.14 | 64.03 | 77.37 | 18.06 | 62.06 | 42.97 | 35.81 | 61.64 | 59.85 |
| TPT [21] | 91.12 | 84.82 | 62.38 | 63.34 | 78.53 | 18.93 | 63.65 | 43.62 | 35.38 | 62.62 | 60.44 |
| C-TPT [24] | 92.33 | 85.23 | 60.76 | 66.02 | 78.51 | 18.06 | 63.59 | 45.09 | 34.64 | 63.07 | 60.73 |
| RLCF [39] | 91.28 | 84.44 | 61.01 | 61.63 | 78.49 | 17.40 | 63.41 | 42.61 | 35.93 | 63.42 | 59.96 |
| MTA [40] | 91.97 | 86.29 | 63.38 | 64.35 | 79.07 | 20.19 | 64.32 | 43.79 | 34.57 | 63.34 | 61.13 |
| ZERO [41] | 91.81 | 85.88 | 62.49 | 62.57 | 78.51 | 20.07 | 64.33 | 42.61 | 32.05 | 62.97 | 60.33 |
| TTL [42] | 91.48 | 85.94 | 62.32 | 62.97 | 78.54 | 18.54 | 64.03 | 43.79 | 32.36 | 62.89 | 60.29 |
| TPS [43] | 91.64 | 86.10 | 63.00 | 63.78 | 79.00 | 19.71 | 63.75 | 43.56 | 35.07 | 63.42 | 60.90 |
| R-TPT [25] | 90.75 | 85.17 | 63.10 | 62.12 | 78.69 | 19.53 | 64.16 | 42.49 | 31.86 | 63.10 | 60.10 |
| TDA [26] | 92.09 | 85.36 | 61.07 | 66.02 | 78.21 | 19.32 | 64.05 | 44.80 | 42.53 | 64.45 | 61.79 |
| DMN$^W$ [44] | 90.75 | 85.64 | 61.36 | 65.57 | 78.11 | 18.60 | 63.32 | 43.91 | 43.56 | 63.68 | 61.45 |
| DMN [44] | 91.03 | 86.73 | 63.60 | 64.51 | 79.37 | 20.37 | 65.22 | 45.09 | 36.95 | 65.05 | 61.79 |
| OnZeta [45] | 91.28 | 86.48 | 61.02 | 65.08 | 78.88 | 18.66 | 63.09 | 45.21 | 40.95 | 63.57 | 61.42 |
| BoostAdapter [46] | 91.93 | 85.88 | 63.67 | 65.81 | 79.52 | 21.15 | 64.96 | 45.39 | 42.23 | 65.16 | 62.57 |
| DPE [47] | 92.74 | 86.10 | 61.42 | 64.88 | 77.86 | 17.52 | 64.02 | 47.04 | 34.85 | 65.85 | 61.23 |
| ECALP [27] | 92.58 | 86.43 | 61.21 | 68.57 | 79.95 | 20.04 | 65.76 | 46.22 | 47.85 | 66.48 | 63.51 |
| DynaPrompt [28] | 92.21 | 85.99 | 61.92 | 65.00 | 78.98 | 17.85 | 63.50 | 43.50 | 34.86 | 62.81 | 60.66 |

Table 17: Accuracies (%) of TTA methods on **ImageNet-X datasets** with CLIP-ViT-B/32.

| | ImageNet | ImageNet-V2 | ImageNet-R | ImageNet-A | ImageNet-Sketch | Avg. | OOD Avg. |
|---|---|---|---|---|---|---|---|
| CLIP [1] | 62.04 | 54.77 | 66.23 | 29.53 | 40.84 | 50.68 | 47.84 |
| TPT [21] | 63.47 | 56.85 | 69.01 | 34.68 | 41.67 | 53.14 | 50.55 |
| C-TPT [24] | 63.85 | 56.42 | 68.45 | 32.43 | 42.18 | 52.67 | 49.87 |
| RLCF [39] | 63.64 | 57.38 | 70.45 | 37.49 | 42.75 | 54.34 | 52.02 |
| MTA [40] | 65.02 | 58.21 | 70.37 | 38.03 | 43.45 | 55.02 | 52.52 |
| ZERO [41] | 65.29 | 58.91 | 70.76 | 40.48 | 43.65 | 55.82 | 53.45 |
| TTL [42] | 64.99 | 58.91 | 71.16 | 39.65 | 43.84 | 55.71 | 53.39 |
| TPS [43] | 64.77 | 57.75 | 70.21 | 38.61 | 43.14 | 54.90 | 52.43 |
| R-TPT [25] | 64.25 | 57.96 | 69.94 | 36.61 | 41.60 | 54.07 | 51.53 |
| TDA [26] | 63.50 | 55.45 | 67.44 | 31.23 | 43.01 | 52.13 | 49.28 |
| DMN$^W$ [44] | 62.68 | 54.76 | 66.12 | 29.60 | 42.37 | 51.11 | 48.21 |
| DMN [44] | 65.72 | 58.98 | 70.36 | 41.52 | 44.84 | 56.28 | 53.93 |
| BoostAdapter [46] | 65.01 | 56.87 | 70.39 | 38.99 | 44.40 | 55.13 | 52.66 |
| ECALP [27] | 64.39 | 55.43 | 67.66 | 29.96 | 44.71 | 52.43 | 49.44 |
| DynaPrompt [28] | 63.92 | 56.47 | 68.94 | 32.99 | 42.06 | 52.88 | 50.12 |

Table 18: Accuracies (%) of TTA methods on **fine-grained datasets** with **SigLIP** (ViT-B/16).

| | Caltech101 | Pets | Cars | Flowers | Food101 | Aircraft | SUN397 | DTD | EuroSAT | UCF101 | Avg. |
|---|---|---|---|---|---|---|---|---|---|---|---|
| SigLIP [4] | 97.93 | 93.13 | 90.70 | 84.37 | 87.30 | 40.68 | 69.64 | 63.00 | 41.33 | 70.79 | 73.89 |
| TPT [21] | 98.05 | 92.78 | 91.28 | 84.73 | 87.50 | 40.71 | 70.03 | 64.18 | 40.44 | 71.48 | 74.12 |
| C-TPT [24] | 97.97 | 92.94 | 90.77 | 84.17 | 87.37 | 40.05 | 69.64 | 63.59 | 40.62 | 70.71 | 73.78 |
| RLCF [39] | 96.47 | 92.20 | 86.30 | 81.20 | 85.15 | 33.72 | 63.97 | 54.02 | 30.09 | 66.24 | 68.94 |
| MTA [40] | 98.05 | 92.94 | 91.52 | 84.86 | 87.29 | 39.78 | 69.80 | 65.19 | 33.56 | 71.32 | 73.43 |
| ZERO [41] | 97.65 | 92.26 | 90.98 | 84.29 | 86.54 | 34.05 | 69.03 | 64.07 | 24.48 | 70.76 | 71.41 |
| TTL [42] | 97.97 | 92.53 | 91.12 | 84.17 | 87.54 | 38.76 | 70.12 | 65.25 | 37.22 | 72.14 | 73.68 |
| TPS [43] | 98.05 | 92.53 | 91.37 | 85.10 | 87.48 | 39.96 | 69.60 | 65.13 | 34.01 | 72.38 | 73.56 |
| R-TPT [25] | 97.32 | 92.94 | 90.72 | 84.94 | 87.02 | 32.82 | 69.09 | 64.66 | 26.37 | 70.18 | 71.61 |

Table 19: Expected calibration error (%) of TTA methods on **fine-grained datasets** with CLIP-ResNet50.

| | Caltech101 | Pets | Cars | Flowers | Food101 | Aircraft | SUN397 | DTD | EuroSAT | UCF101 | ECE Avg. ↓ |
|---|---|---|---|---|---|---|---|---|---|---|---|
| CLIP [1] | 4.50 | 5.62 | 4.47 | 2.96 | 2.69 | 6.31 | 3.82 | 8.82 | 14.76 | 3.01 | 5.70 |
| TPT [21] | 3.98 | 3.87 | 4.16 | 13.51 | 5.15 | 15.97 | 9.03 | 25.22 | 21.17 | 10.89 | 11.30 |
| C-TPT [24] | 3.10 | 3.25 | 1.81 | 4.09 | 1.80 | 11.12 | 2.81 | 21.55 | 13.62 | 2.96 | 6.61 |
| RLCF [39] | 6.00 | 8.51 | 11.44 | 24.47 | 13.70 | 23.56 | 20.21 | 37.36 | 35.95 | 21.43 | 20.26 |
| MTA [40] | 4.63 | 4.99 | 5.97 | 13.68 | 11.55 | 7.70 | 10.43 | 22.04 | 26.50 | 14.49 | 12.20 |
| TPS [43] | 7.14 | 9.63 | 17.51 | 22.77 | 15.74 | 23.16 | 20.20 | 35.19 | 39.18 | 21.08 | 21.16 |
| R-TPT [25] | 6.48 | 4.27 | 1.63 | 12.27 | 5.44 | 12.42 | 7.88 | 25.00 | 29.44 | 8.32 | 11.32 |
| TDA [26] | 5.83 | 3.00 | 1.54 | 5.68 | 2.40 | 15.74 | 5.80 | 19.19 | 23.56 | 9.36 | 9.21 |
| DMN$^W$ [44] | 5.22 | 4.43 | 14.86 | 9.79 | 6.56 | 31.75 | 16.39 | 36.78 | 23.05 | 16.46 | 16.53 |
| DMN [44] | 3.59 | 5.93 | 9.66 | 10.44 | 4.55 | 26.81 | 12.83 | 27.35 | 13.63 | 14.14 | 12.89 |
| BoostAdapter [46] | 10.27 | 2.56 | 16.15 | 6.72 | 14.83 | 48.03 | 18.03 | 22.20 | 23.03 | 10.67 | 17.25 |
| ECALP [27] | 11.46 | 14.35 | 39.35 | 33.79 | 22.16 | 73.12 | 31.45 | 51.65 | 12.18 | 32.55 | 32.21 |
| DynaPrompt [28] | 2.81 | 2.41 | 1.09 | 4.97 | 2.35 | 9.97 | 3.05 | 17.46 | 22.89 | 7.21 | 7.42 |

Table 20: Accuracies (%) of Episodic TTA methods on **fine-grained datasets** under the invasion of OOD samples with CLIP-ResNet50.

| | Caltech101 | Pets | Cars | Flowers | Food101 | Aircraft | SUN397 | DTD | EuroSAT | UCF101 | Avg. |
|---|---|---|---|---|---|---|---|---|---|---|---|
| CLIP [1] | 90.58 | 84.91 | 55.42 | 64.96 | 78.70 | 17.35 | 66.64 | 51.93 | 38.74 | 65.01 | 61.42 |
| TPT [21] | 93.03 | 85.80 | 58.55 | 64.86 | 79.51 | 18.37 | 69.31 | 50.48 | 32.41 | 65.65 | 61.80 |
| C-TPT [24] | 92.45 | 84.06 | 55.65 | 68.47 | 79.45 | 18.43 | 68.86 | 53.14 | 39.52 | 65.54 | 62.56 |
| RLCF [39] | 92.83 | 84.79 | 57.87 | 62.30 | 79.32 | 17.71 | 68.65 | 50.60 | 30.45 | 65.96 | 61.05 |
| MTA [40] | 92.38 | 85.63 | 57.92 | 63.34 | 79.17 | 20.05 | 68.58 | 51.21 | 31.12 | 66.23 | 61.56 |
| ZERO [41] | 91.93 | 85.47 | 57.92 | 61.82 | 77.56 | 19.63 | 68.51 | 48.67 | 23.26 | 64.80 | 59.96 |
| TPS [43] | 92.64 | 85.75 | 57.90 | 63.53 | 79.29 | 19.81 | 68.41 | 52.66 | 28.69 | 65.81 | 61.45 |
| R-TPT [25] | 91.67 | 85.19 | 58.15 | 63.34 | 78.51 | 19.03 | 68.85 | 49.28 | 27.21 | 64.59 | 60.58 |

Table 21: Accuracies (%) of online TTA methods on **fine-grained datasets** with CLIP-ResNet50 when OOD samples are mixed into the test data stream.

| | Caltech101 | | Pets | | Cars | | Flowers | | Food101 | | Aircraft | |
|---|---|---|---|---|---|---|---|---|---|---|---|---|
| | w/ OOD | w OOD | w/ OOD | w OOD | w/ OOD | w OOD | w/ OOD | w OOD | w/ OOD | w OOD | w/ OOD | w OOD |
| CLIP | 90.70 | 90.70 | 84.85 | 84.85 | 55.40 | 55.40 | 64.96 | 64.96 | 78.65 | 78.65 | 17.53 | 17.53 |
| TDA [26] | 92.77 | 92.25 | 85.58 | 85.69 | 56.92 | 56.92 | 68.47 | 68.09 | 79.40 | 79.37 | 17.77 | 17.53 |
| DMN$^W$ [44] | 89.61 | 89.93 | 85.69 | 85.69 | 56.27 | 55.95 | 66.10 | 68.00 | 78.97 | 78.88 | 18.07 | 17.83 |
| DMN [44] | 90.83 | 90.96 | 85.75 | 86.48 | 59.15 | 58.45 | 64.86 | 65.91 | 78.78 | 78.75 | 19.75 | 19.15 |
| OnZeta [45] | 90.51 | 90.32 | 87.88 | 83.16 | 57.67 | 54.72 | 61.54 | 62.39 | 80.04 | 78.55 | 20.11 | 17.05 |
| BoostAdapter [46] | 91.54 | 91.74 | 85.97 | 85.69 | 58.15 | 58.60 | 67.71 | 67.52 | 79.73 | 79.74 | 19.69 | 19.93 |
| DPE [47] | 91.67 | 91.09 | 68.01 | 65.66 | 47.43 | 42.05 | 66.38 | 66.57 | 79.23 | 79.13 | 17.47 | 13.99 |
| ECALP [27] | 93.09 | 92.70 | 87.21 | 87.37 | 59.02 | 59.00 | 68.66 | 68.76 | 80.41 | 80.38 | 18.19 | 18.19 |
| DynaPrompt [28] | 92.58 | 92.77 | 85.41 | 85.35 | 57.25 | 57.50 | 65.43 | 64.20 | 79.83 | 79.47 | 17.41 | 17.59 |

| | SUN397 | | DTD | | EuroSAT | | UCF101 | | Avg. | | |
|---|---|---|---|---|---|---|---|---|---|---|---|
| | w/ OOD | w OOD | w/ OOD | w OOD | w/ OOD | w OOD | w/ OOD | w OOD | w/ OOD | w OOD | Δ |
| CLIP | 66.64 | 66.64 | 51.69 | 51.69 | 38.76 | 38.76 | 65.01 | 65.01 | 61.42 | 61.42 | (− 0.00) |
| TDA [26] | 69.12 | 68.69 | 53.50 | 52.05 | 44.43 | 44.55 | 67.12 | 66.44 | 63.51 | 63.16 | (↓ 0.35) |
| DMN$^W$ [44] | 67.39 | 67.25 | 52.78 | 53.50 | 44.10 | 40.67 | 66.70 | 66.97 | 62.57 | 62.47 | (↓ 0.10) |
| DMN [44] | 69.40 | 69.38 | 50.12 | 50.60 | 38.50 | 38.48 | 66.54 | 66.54 | 62.37 | 62.47 | (↑ 0.10) |
| OnZeta [45] | 69.31 | 68.00 | 48.67 | 48.67 | 43.69 | 44.79 | 66.91 | 66.65 | 62.63 | 61.43 | (↓ 1.20) |
| BoostAdapter [46] | 69.15 | 69.12 | 53.14 | 52.90 | 49.14 | 48.43 | 67.49 | 66.75 | 64.17 | 64.04 | (↓ 0.13) |
| DPE [47] | 68.78 | 68.19 | 52.17 | 45.53 | 41.95 | 39.98 | 67.81 | 67.97 | 60.09 | 58.02 | (↓ 2.07) |
| ECALP [27] | 70.69 | 69.83 | 56.04 | 54.95 | 42.86 | 41.98 | 70.45 | 70.40 | 64.66 | 64.36 | (↓ 0.31) |
| DynaPrompt [28] | 68.93 | 68.86 | 52.05 | 51.21 | 32.12 | 32.57 | 65.07 | 64.33 | 61.61 | 61.39 | (↓ 0.22) |

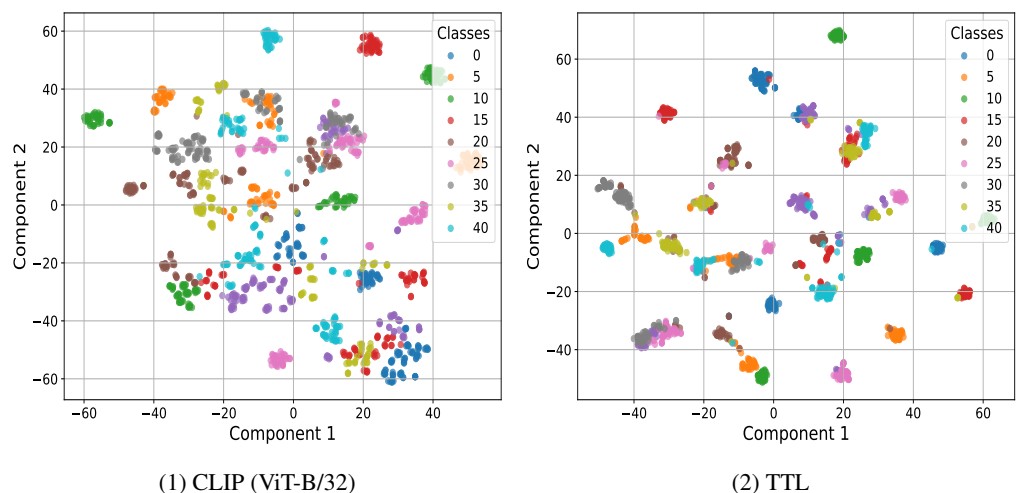

(1) CLIP (ViT-B/32)  (2) TTL

Figure 4: t-SNE visualization of visual features from CLIP and TTL on a subset of the UCF101 dataset.

