# OpenReview forum: "The Illusion of Progress? A Critical Look at Test-Time Adaptation for Vision-Language Models"
_NeurIPS.cc/2025/Datasets_and_Benchmarks_Track — NeurIPS 2025 Datasets and Benchmarks Track poster_

### Official Review · Reviewer_4Juw · 2025-06-08

**Rating:** 4
**Confidence:** 1

**Summary:**

This paper proposes TTA-VLM, containing 8 episodic TTA methods and 7 online TTA methods that focus on more than accuracy, including also robustness, calibration, out-of-distribution detection, and stability. Then, it evaluates them on 15 existing and widely used datasets. The study extends the traditionally tested CLIP models and encompasses more VLMs, and suggests limitations of previous benchmarks.

**Dataset Code Accessibility:**

Yes

**Ethical Considerations:**

No, there are no or only very minor ethics concerns

**Final Justification:**

I thank the authors for answering my questions, and I am happy with the rebuttal. Since I am unfamiliar with the related works, I prefer to maintain my score of borderline accept. I also suggest that the AC down-weights my evaluation when appropriate. Thank you!

**Limitations Weaknesses:**

Unfortunately, the assigned paper is very much out of my domain of knowledge. While I know the basics of TTA and VLMs, I do not feel that I have enough background knowledge to conduct a critical evaluation of this work. I outline two weaknesses based on educated guesses.
* The criticism of the existing benchmark is not carefully justified. I look for insights into why TTA fails in these cases.
* All tasks in the proposed benchmark are classification problems. A wider variety of tasks can make the benchmark more comprehensive.

**Strengths Contributions:**

* The proposed benchmark covers a wide range of datasets, model types, and adaptation paradigms.
* The benchmark tests more than the accuracy metrics and also tailors robustness, calibration, out-of-distribution detection, and stability. Hence, the evaluation is more comprehensive.

---

> ### Author Rebuttal · Authors · 2025-07-31
>
> Thanks for your valuable comments. We are encouraged the variety of metrics and datasets used in our benchmark. Questions are answered in turn, and we hope that the responses can address your concerns.
>
> > Q1: The criticism of the existing benchmark is not carefully justified. I look for insights into why TTA fails in these cases.
>
> **A:** Thank you for your suggestion. Before our work, there was no systematic benchmark specifically designed to evaluate VLM-based TTA methods. Most existing results are reported in the original papers of each method and most works choose to replicate others' results without consistent TTA settings (e.g., prompt design, TTA paradigm). In contrast, our unified framework standardizes these components to eliminate variability caused by different experimental environments and introduces many other metrics and models. This has revealed inconsistencies in performance and, in some cases, unexpected failures of certain algorithms.
>
> As for why TTA fails (i.e., negative transfer) in some scenarios, we identify several possible factors:
>
> 1. **Failing on adversarially-trained model**: Adversarial training significantly alters a model’s gradient landscape. Many existing TTA methods, when applied to these models, disrupt the carefully learned robust feature distributions, which in turn diminishes their resistance to input perturbations.
> 2. **Failing with adversarial examples**: TTA methods are typically designed to refine wrong predictions for samples near the decision boundary. However, adversarial examples often lie far from the true class distribution. As a result, non-ensemble-based TTA methods perform poorly, sometimes showing near-zero effectiveness.
> 3. **Failing on general datasets**: Some TTA methods perform well in original papers due to specific prompt templates or prompt ensembling strategies tailored to the task. Our unified benchmark removes such biases by using the same VLMs and a standardized prompt setup, which leads to noticeable differences in clean-data performance.
>
>
> > Q2: All tasks in the proposed benchmark are classification problems. A wider variety of tasks can make the benchmark more comprehensive.
>
> **A:** Thank you for highlighting this point. Our current benchmark is indeed focused on classification tasks using vision-language models. This is primarily because most existing TTA algorithms, with the exception of RLCF, are specifically designed for classification problems. Although TTA methods for other tasks (e.g., segmentation) have been proposed, they are relatively recent, limited in number, and often rely on different experimental setups, making it difficult to include them in a unified benchmark alongside classification-focused methods.
>
> We will incorporate a discussion of those methods for other tasks (as shown in the following table) and cite relevant work in the revision. We believe that our benchmark framework can still inspire experimental settings and help guide the development of TTA approaches for a broader range of tasks.
>
> | Method    | TTA paradigm |       Task |
> |------------|-------------|---------------------------|
> | AnoCLIP [1]    | episodic TTA |       Anomaly Localization |
> | T3AL  [2]     | episodic TTA |        Action Localization |
> | DTS-TPT [3]  | episodic TTA |       Activity Recognition |
> | CLIP-DIY [4] | episodic TTA |      Semantic Segmentation |
> | CLIPtrase [5] | episodic TTA |      Semantic Segmentation |
> | VocAda [6]  | episodic TTA |           Object Detection |
> | TTCS   [7] |   online TTA | Medical Image Segmentation |
> | TEST-V [8] |   online TTA |       Video Classification |
>
> [1] Deng et al. "Bootstrap fine-grained vision-language alignment for unified zero-shot anomaly localization." arXiv 2023.
>
> [2] Liberatori et al. "Test-time zero-shot temporal action localization." CVPR 2024.
>
> [3] Yan et al. "DTS-TPT: dual temporal-sync test-time prompt tuning for zero-shot activity recognition." IJCAI 2024.
>
> [4] Wysoczańska et al. "Clip-diy: Clip dense inference yields open-vocabulary semantic segmentation for-free." WACV 2024.
>
> [5] Shao et al. "Explore the potential of clip for training-free open vocabulary semantic segmentation." ECCV 2024.
>
> [6] Liu et al. "Test-time vocabulary adaptation for language-driven object detection." arXiv 2025.
>
> [7] Chen et al. "Test-time medical image segmentation using clip-guided sam adaptation." BIBM 2024.
>
> [8] Yan et al. "TEST-V: test-time support-set tuning for zero-shot video classification." arXiv 2025.

---

### Official Review · Reviewer_dwCD · 2025-06-28

**Rating:** 5
**Confidence:** 5

**Summary:**

This paper systematically studies the test-time adaptation performance of existing vision-language models, and proposes a TTA-VLM benchmark.

This benchmark tests various TTA and online TTA methods on 15 datasets.

Through extensive experiments, some valuable findings for existing TTA methods are reported.

**Dataset Code Accessibility:**

Yes

**Ethical Considerations:**

No, there are no or only very minor ethics concerns

**Final Justification:**

My concerns have been addressed.

I would like to improve my score from borderline accept to accept.

Hope the authors could take the comments when revising the camera-ready version.

**Limitations Weaknesses:**

- The benchmarked models are not very new or comprehensive. More advanced models, for example Vision Mamba and more advanced transformer models, should be tested.

- This paper lacks an in-depth feature space analysis between the compared methods on the benchmark, which is necessary to demonstrate the corresponding challenges of the proposed benchmark.

- A more comprehensive table comparison, especially in terms of the sample number and dataset number, between the proposed benchmark and the existing benchmark, should be provided. It is very meaningful to highlight the contribution of this benchmark over existing benchmarks.

- Some visual examples on the benchmark should be provided.

- Some typos and grammar mistakes should be corrected.

- The reference section is not well organized. The page number of each paper is missing, and please use the full name of each conference or journal.

**Strengths Contributions:**

+ Overall this paper is well-written and easy-to-follow.

+ Test-time adaptation is a practical setting for deep learning models, and deserves a systematic benchmark.

+ The experiments and method comparison are systematic.

---

> ### Author Rebuttal · Authors · 2025-07-31
>
> Thanks for your valuable comments. We are encouraged that the reviewer appreciate systematic experiments of our benchmark. Questions are answered in turn, and we hope that the responses can address your concerns.
>
> > Q1: More advanced models, for example Vision Mamba and more advanced transformer models, should be tested.
>
> **A:** Thank you for the insightful suggestion! We agree that evaluating TTA methods on a broader range of model architectures would provide a more comprehensive assessment of its generalization.
>
> Our current benchmark primarily targets vision-language models such as CLIP, and thus does not include purely visual models like VisionMamba. To address your concern and better demonstrate the versatility of TTA methods, we conduct preliminary experiments on two visual models: ViT-base and VisionMamba-base, both pre-trained on ImageNet. These experiments were performed using the ImageNet test set.
>
> To adapt TTA methods to visual models, we modified the prompt optimization strategy by optimizing NormLayer. Since experiments are still in progress, the table below presents average accuracy results over the first 15,000 samples of the test set. We will provide complete results later.
>
> **ViT-base (ImageNet 15,000 samples)**
> | Method | Top-1 ACC. |
> |-------------|--------------|
> | ViT-B/16| 84.80 |
> | +TPT   | 85.99 |
> | +C-TPT | 85.99 |
> | +MTA   | 86.92 |
> | +ZERO  | 87.17 |
> | +TPS   | 86.19 |
>
> **VisionMamba-base (ImageNet 15,000 samples)**
> | Method | Top-1 ACC. |
> |-------------|--------------|
> | ViM-base | 84.20 |
> | +TPT   | 84.47 |
> | +C-TPT | 84.47 |
> | +MTA   | 83.42 |
> | +ZERO  | 86.09 |
> | +TPS   | 85.35 |
>
> As shown, the TTA methods lead to performance improvements across both visual models, with ZERO consistently yielding the highest accuracy. This suggests that voting-based strategies generalize well across architectures.
>
>
> > Q2: This paper lacks an in-depth feature space analysis between the compared methods on the benchmark.
>
> **A:** Thanks for your valuable comments. Visualizing the feature space can help illustrate how TTA methods adapt to specific tasks, thereby providing indirect evidence of their effectiveness. However, due to the different optimization spaces employed by carious TTA methods, it is challenging to analyze them within a unified framework. Thus, we propose visualizing features separately within the visual and textual embedding spaces:
>
> 1. **For methods that adapt the visual branch (e.g., TTL), we propose visualizing the distribution of the adapted visual features.** The visualization results show that, compared to CLIP, TTL produces image features that are more clearly clustered by category, which correlates with its improved accuracy.
>
> 2. **For methods that optimize the textual branch (e.g., TPT and C-TPT), we propose visualize the adapted text features to analyze the semantic relationships between categories.** The visualization results indicate that the text features produced by C-TPT are more dispersed compared to those of methods like TPT. This increased dispersion enhances the separability of similar categories and contributes to improved calibration.
>
> Since we can not provide figures during the rebuttal process, we will include the above discussion in the revision.
>
> > Q3: A more comprehensive table comparison, especially in terms of the sample number and dataset number, between the proposed benchmark and the existing benchmark, should be provided.
>
> **A:** Thank you for your suggestion. We would like to clarify that, to the best of our knowledge, no prior work has comprehensively or fairly evaluated VLM-based TTA methods. Therefore, we did not include a comparison table of datasets and sample sizes with other benchmarks.
>
> We use the same datasets in our proposed benchmark as the existing TTA work. Also, we report the sample size, number of classes, and image content in **Table 2** of the supplementary material.
>
> While using the same dataset, our benchmark introduces several novel aspects: the additional VLM, the collaboration with training-time methods, and the use of multiple evaluation metrics for TTA. These design choices represent the core contributions of our work.
>
> > Q4: Some visual examples on the benchmark should be provided.
>
> **A:** Great idea. We will add visual examples in the revision to show the differences in style and content between the datasets.
>
> > Q5: Some typos and grammar mistakes should be corrected.
>
> **A:** We will check carefully and correct those mistakes in the revision.
>
> > Q6: The reference section is not well organized.
>
> **A:** Thanks for pointing that. Since some conference papers do not have page numbers, we choose to omit the page numbers in the submission for the sake of uniform formatting.
>
> We will improve our paper and organize the reference section according to the official citation entry.

---

> > ### Comment · Reviewer_dwCD · 2025-08-05
> > **Re: Rebuttal by Authors**
> >
> > Dear Authors
> >
> > My concerns have been addressed.
> >
> > I would like to improve my score from borderline accept to accept.
> >
> > Hope the authors could take the comments when revising the camera-ready version.
> >
> > Reviewer

---

> > > ### Author Response · Authors · 2025-08-05
> > >
> > > Thank you for appreciating our work and for raising your score.
> > >
> > > We are glad that our response addressed your concerns effectively.
> > >
> > > Your support always means a lot to us!

---

### Official Review · Reviewer_nJYJ · 2025-07-03

**Rating:** 4
**Confidence:** 5

**Summary:**

This paper presents TTA-VLM, a benchmark designed to evaluate test-time adaptation (TTA) methods for vision-language models (VLMs). The authors implement and unify 15 recent TTA methods (both episodic and online), covering 15 datasets and two backbone models (CLIP and SigLIP), and examine how well these methods work in terms of accuracy, robustness, calibration, and OOD detection. They also explore how TTA interacts with training-time tuning approaches like CoOp, MaPLe, and TeCoA. The findings are sobering: despite growing attention to TTA, gains are often marginal and can come at the cost of reliability.

**Additional Feedback:**

This paper is a good contribution to the Benchmarks Track. It doesn’t introduce a new TTA method, but it does something more valuable at this point: it brings order and clarity to a rapidly growing subfield that’s in need of both. The benchmark is carefully designed, the findings are significant, and the code release makes it likely to have a good impact.

**Dataset Code Accessibility:**

Yes

**Dataset Code Comments:**

The authors provide a comprehensive and well-documented codebase for the TTA-VLM benchmark at https://github.com/TomSheng21/tta-vlm. The repository includes implementations all these test-time adaptation methods, standardized evaluation protocols, and detailed instructions for reproducing all experiments.

**Ethical Considerations:**

No, there are no or only very minor ethics concerns

**Final Justification:**

Upon thorough evaluation of the authors' rebuttal and the peer-review discourse, I maintain my substantive concern regarding the title "TTA for VLM", as it presents an unwarranted generalization. The scope of this work is rigorously confined to image classification—merely one facet of VLM applications—yet the title misleadingly suggests comprehensive coverage of TTA methodologies across VLMs. While the convention of employing expansive titles (e.g., "XXX for VLM") for narrow technical contributions is regrettably prevalent in the literature, this practice becomes particularly concerning when establishing purported benchmarks, as it may engender false expectations regarding the work's generalizability.

Nevertheless, the manuscript undeniably offers a methodical comparative analysis of TTA approaches specifically for VLM-based image classification, delivering valuable empirical findings for this niche domain. Therefore, I incline to keep my initial evaluation.

**Limitations Weaknesses:**

1. The benchmark is limited to classification, which feels a bit narrow given the broader scope of VLMs. It would be great to see at least one task from a different family (e.g., retrieval or VQA), or at least a discussion of what extending TTA to such tasks might involve.
2. The short experiment on using multiple text prompts is intriguing, especially the gains seen with certain methods. Given how central prompts are to VLMs, it seems like this deserves more space.
3. TTA methods are used at inference time, often in latency-sensitive settings. A brief comparison of computational costs would add real value for practitioners.

**Strengths Contributions:**

1. TTA for VLMs is a fast-emerging area, and the current literature is somewhat chaotic—papers use different setups, inconsistent baselines, and rarely look beyond accuracy. This benchmark is addressing a real gap in the community.
2. The authors make a commendable effort to standardize all the variables that usually confound comparison—augmentations, prompts, hyperparams—and they report results across multiple models and datasets. This is the kind of rigor that is often missing in TTA papers.
3. The inclusion of calibration, robustness, and OOD detection is especially welcome. It’s one thing to improve accuracy, another to make the model more trustworthy. The results clearly show that many TTA methods fail to strike that balance.
4. The conclusion that many TTA methods don’t meaningfully outperform early baselines (like TPT) under fair conditions is an important message. So is the observation that TTA often struggles to work with fine-tuned models.
5. The authors release their code and models, making it easy for others to build on this work. That’s exactly the kind of contribution that fits the Benchmarks Track.

---

> ### Author Rebuttal · Authors · 2025-07-31
>
> Thanks for your valuable comments. We are encouraged that the reviewer appreciate the observations from our benchmark. Questions are answered in turn, and we hope that the responses can address your concerns.
>
> > Q1: It would be great to see at least one task from a different family (e.g., retrieval or VQA), or at least a discussion of what extending TTA to such tasks might involve.
>
> **A:**  Thank you for highlighting this point. Our current benchmark is indeed focused on classification tasks using vision-language models. This is primarily because most existing TTA algorithms, with the exception of RLCF, are specifically designed for classification problems. Although TTA methods for other tasks (e.g., segmentation) have been proposed, they are relatively recent, limited in number, and often rely on different experimental setups, making it difficult to include them in a unified benchmark alongside classification-focused methods.
>
> We will incorporate a discussion of those methods for other tasks (as shown in the following table) and cite relevant work in the revision. We believe that our benchmark framework can still inspire experimental settings and help guide the development of TTA approaches for a broader range of tasks.
>
> | Method    | TTA paradigm |       Task |
> |------------|-------------|---------------------------|
> | AnoCLIP [1]    | episodic TTA |       Anomaly Localization |
> | T3AL  [2]     | episodic TTA |        Action Localization |
> | DTS-TPT [3]  | episodic TTA |       Activity Recognition |
> | CLIP-DIY [4] | episodic TTA |      Semantic Segmentation |
> | CLIPtrase [5] | episodic TTA |      Semantic Segmentation |
> | VocAda [6]  | episodic TTA |           Object Detection |
> | TTCS   [7] |   online TTA | Medical Image Segmentation |
> | TEST-V [8] |   online TTA |       Video Classification |
>
> [1] Deng et al. "Bootstrap fine-grained vision-language alignment for unified zero-shot anomaly localization." arXiv 2023.
>
> [2] Liberatori et al. "Test-time zero-shot temporal action localization." CVPR 2024.
>
> [3] Yan et al. "DTS-TPT: dual temporal-sync test-time prompt tuning for zero-shot activity recognition." IJCAI 2024.
>
> [4] Wysoczańska et al. "Clip-diy: Clip dense inference yields open-vocabulary semantic segmentation for-free." WACV 2024.
>
> [5] Shao et al. "Explore the potential of clip for training-free open vocabulary semantic segmentation." ECCV 2024.
>
> [6] Liu et al. "Test-time vocabulary adaptation for language-driven object detection." arXiv 2025.
>
> [7] Chen et al. "Test-time medical image segmentation using clip-guided sam adaptation." BIBM 2024.
>
> [8] Yan et al. "TEST-V: test-time support-set tuning for zero-shot video classification." arXiv 2025.
>
> > Q2: The short experiment on using multiple text prompts is intriguing, especially the gains seen with certain methods. Given how central prompts are to VLMs, it seems like this deserves more space.
>
> **A:** Thank you for your insightful comment. While most experiments in our paper use the default prompt template ("a photo of a {class}"), we agree that prompt design plays a critical role in the performance of VLMs. We will discuss the importance of prompt from two aspects:
>
> 1. **Multiple Templates**: Our submission contains this part in Sec.3. Some TTA methods that do not optimize textual prompts can use multiple templates for initialization. Many of these methods benefit from using the introduction of multiple templates.
> 2. **Single Accurate Template**: We report classification accuracy on two fine-grained datasets using different single templates for TTA methods in the following table. Templates that offer more accurate or descriptive class representations tend to enhance the zero-shot performance of VLMs and, consequently, improve the effectiveness of TTA methods. However, better prompts may reduce the room for improvement by episodic TTA algorithms, as the base predictions are already stronger. Still, most online TTA methods continue to yield notable gains even with strong initial prompts. Specifically, different methods have different gains when using better templates. For instance, on the EuroSAT dataset, RLCF and ECALP achieve the highest performance within their respective paradigms when the enhanced template is applied. However, both of them underperform compared to TPT and DMN when using the default template.
>
> | Dataset | Flowers102 | Flowers102 | EuroSAT | EuroSAT |
> |-|-|-|-|-|
> | template | a photo of a | a flower photo of a | a photo of a | a centered satellite photo of a |
> | CLIP-RN50    | 61.67 | 65.73 | 23.68 | 33.80 |
> | Episodic TTA | | | | |
> | TPT          | 62.08 | 65.41 | **28.41** | 36.40 |
> | C-TPT        | **64.80** | **66.42** | 26.98 | 31.77 |
> | RLCF         | 59.16 | 64.07 | 27.23 | **37.17** |
> | MTA          | 61.02 | 64.96 | 22.53 | 33.96 |
> | TPS          | 61.55 | 65.57 | 24.30 | 35.84 |
> | R-TPT        | 61.35 | 63.62 | 21.40 | 29.30 |
> | Online TTA | | | | |
> | TDA          | 65.16 | 67.52 | 31.15 | 41.21 |
> | DMN          | 64.11 | 67.44 | **37.75** | 44.17 |
> | OnZeta       | 60.94 | 64.39 | 30.53 | 41.65 |
> | BoostAdapter | 65.08 | 67.64 | 32.51 | 40.53 |
> | DPE          | 63.82 | 67.88 | 25.85 | 34.17 |
> | ECALP        | **66.18** | **67.93** | 30.22 | **48.37** |
>
> We will include the above discussion in the revision.
>
> > Q3: A brief comparison of computational costs would add real value for practitioners.
>
> **A:** We agree that computational costs is critical for the practical deployment of TTA algorithms. The overall cost mainly includes **runtime and GPU memory consumption**. To address this, we provide a comparative analysis of various TTA algorithms on the ImageNet dataset, as shown in **Table 3** of the supplementary material.
>
> It's worth noting that the computational cost can be further reduced in downstream tasks with fewer classes or when memory-saving strategies are employed. Nevertheless, we believe the relative trends between algorithms remain consistent across such variations.

---

> > ### Author Response · Authors · 2025-08-05
> >
> > Dear Reviewer nJYJ,
> >
> > Thank you again for the great efforts and valuable comments. We have carefully addressed each of your comments in our point-by-point response. We hope you might find the response satisfactory.
> >
> > We value the reviewer-author discussion phase and are very much **looking forward to hearing from you about any further feedback**.
> >
> > Best, Authors

---

> > ### Comment · Reviewer_nJYJ · 2025-08-07
> > **Futher Comments**
> >
> > Thank you for your detailed response.
> >
> > The author's responses to points two and three are satisfactory, and their agreement to incorporate these changes is appreciated.
> >
> > Regarding the first point, I appreciate the author's perspective, but I still have some reservations. Given that the title suggests a broader discussion on TTA's design and application in VLMs, I would expect the paper to cover a wider range of tasks rather than focusing primarily on image classification. In its current form, the scope feels somewhat limited.

---

> > > ### Author Response · Authors · 2025-08-07
> > >
> > > Thanks for your positive feedback on our responses to the textual prompts and computational costs.
> > >
> > > In response to your suggestion about covering more tasks, we would like to provide further clarification.
> > >
> > > Most existing TTA papers focus primarily on classification tasks. Works on other tasks (e.g., segmentation, detection) are relatively scarce, making them less suitable for inclusion in the current benchmark. For example, we analyze 48 existing online TTA works and find that 41 of them focus on image classification, while only 2 of them are designed for segmentation.
> > >
> > > We agree on the value of including a broader range of tasks. While they may not yet be suitable for inclusion in benchmarks, we will cite and discuss related works in these tasks to provide readers with a more comprehensive understanding.

---

### Official Review · Reviewer_1qXX · 2025-07-03

**Rating:** 5
**Confidence:** 3

**Summary:**

The paper focuses on proposing a benchmark for a standardized and consistent evaluation of test-time adaptation (TTA) of Vision-Language Models (VLMs).

The authors first highlight current limitations of the experimental setups in TTA works:
* Lack of reproduction of previous works and reliance solely on reported results, which can lead to inconsistent experimental details and unfair comparisons.
* A limited analysis in most works with a focus on accuracy metrics, without looking at other aspects of the model's behavior (e. g. trustworthiness, robustness, ...)
* A lack of diversity in architectures (most work reporting results on CLIP models)
* A lack of understanding of the interaction between post-training finetuning and TTA.

To address these limitations, the authors propose TTA-VLM, a benchmarks that assess 15 TTA methods (8 episodic and 7 online):
* on 15 fine-grained or ImageNet related datasets, that are commonly used in TTA literature
* using CLIP and SigLIP models with different backbone architectures
* looking at accuracy, as well as calibration, robustness, and OOD detection metrics
* and combining TTA techniques with 3 finetuning techniques.

The authors further conduct 3 sets of experiments:
* The first focuses on simply testing TTA methods and comparing them in a fair consistent setup. In this family of experiments, they also considered the impact of using diverse text templates. Through this set of experiments, they show that:
   * Recent TTA approaches offer a small gain over earlier techniques based on prompting
   * Results on CLIP do not generalise to SigLIP models
   * Using multiple template improve the results on CLIP, with the exception of a few online TTA approaches
* The second set of experiments looks at the interaction between finetuning and TTA. These experiments show consistently a poor collaboration with finetuning techniques, with limited improvement from TTA compared to what can be obtained through finetuning.
* The third set of experiments pushes the analysis beyond accuracy, and shows that:
     * TTA methods tend to decrease calibration and OOD detection
     * When exposed to OOD or adversarial sampled during TTA, models tend to lose performance.

**Additional Feedback:**

See section on weaknesses

**Dataset Code Accessibility:**

NA; not applicable to this submission (e.g., no new dataset, benchmark, code, or data provided)

**Dataset Code Comments:**

The paper proposes a benchmark using existing datasets.

**Ethical Comments:**

The paper proposes a benchmark solely based on public datasets.

**Ethical Considerations:**

No, there are no or only very minor ethics concerns

**Final Justification:**

I thank the authors for the detailed answers and additional results. After reading the rebuttal and other reviews, I confirm my initial score and I am inclined towards accepting the paper.

**Limitations Weaknesses:**

* While the paper considers different metrics, I think an important missing one is efficiency (cost and speed of convergence), especially when combining with finetuning techniques. An interesting question for practitioner is whether it would be more efficient to go through a first phase of finetunine, or to go straight to TTA.
* While the paper considers different datasets, they are still limited in diversity, all being image classification dataset. Considering more tasks and modalities would also be interesting. Nevertheless, this is a weakness also highlighted by the authors in the discussion on limitations.
* Related to the previous point, it is unclear how big is the distribution shift between pretraining data and TTA data. Understanding and analyzing this aspect can also improve our understanding of TTA techniques.

**Strengths Contributions:**

* The proposed benchmark addresses a clearly identified gap in the literature and current practices, and offers a playground that can support the development on improved TTA methods. It also advocate for more standardized and consistent practices, that has the potential push the boundaries of the field and contribute to better and more impactful algorithms.
* The paper and results presentation is really well structured and clear, and put the main takeways of the paper in evidence. This makes the paper easy and enjoyable to read.
* The proposed tests and experiments are interesting, well designed and well justified.
* The combination of all the previous points led to highlighting important limitations and shortcomings of current TTA methods.
* I really appreciated the inclusion of metrics beyond accuracy.

---

> ### Author Rebuttal · Authors · 2025-07-31
>
> Thank you for your valuable comments. We are encouraged that the reviewer appreciate our benchmark setting about metrics and observations from the experiments. Below, we respond to your questions in turn, and we hope our responses address your concerns effectively.
>
> > Q1: An important missing one is efficiency (cost and speed of convergence), especially when combining with finetuning techniques.
>
> **A:** We agree that efficiency is critical for the practical deployment of TTA algorithms. The overall cost mainly includes **runtime and GPU memory consumption**. To address this, we provide a comparative analysis of various TTA algorithms on the ImageNet dataset, as shown in **Table 3** of the supplementary material.
>
> It's worth noting that the computational cost can be further reduced in downstream tasks with fewer classes or when memory-saving strategies are employed. Nevertheless, we believe the relative trends between algorithms remain consistent across such variations.
>
> Regarding convergence speed, most optimization-based TTA methods (e.g., TPT, C-TPT) adopt a single-step optimization strategy, rather than iterative multi-epoch training. Therefore, traditional convergence analyses are not directly applicable.
>
> However, **we can discuss the convergence of TTA methods by analyzing performance across multiple optimization steps**. The following table presents the results of optimization-based algorithm applied with multi-step optimization on four fine-grained datasets. The results indicate that, in most tasks, increasing the number of optimization steps does not bring significant performance improvement. Given the computational overhead associated with multi-step optimization, these findings suggest that the TTA methods effectively converge within a single step.
>
> | 1->2->5 steps | Caltech101          | Flowers102          | DTD                 | UCF101              |
> |-------------|---------------------|---------------------|---------------------|---------------------|
> |CLIP (0 step) | 85.88 | 61.67 | 40.43 | 58.90 |
> | TPT         | 87.91->87.99->87.38 | 62.08->61.18->59.60 | 42.43->41.13->40.72 | 60.64->60.90->60.32 |
> | C-TPT       | 87.75->88.11->87.78 | 64.80->65.24->64.92 | 41.49->41.07->41.66 | 60.16->60.37->61.45 |
> | TPS         | 86.69->87.34->87.30 | 61.55->60.53->59.43 | 40.43->40.72->40.24 | 60.61->60.56->60.26 |
> | R-TPT       | 86.33->86.65->87.05 | 61.35->60.45->60.37 | 41.55->40.07->40.24 | 59.50->60.29->60.19 |
>
>
>
> > Q2: An interesting question for practitioner is whether it would be more efficient to go through a first phase of finetunine, or to go straight to TTA.
>
> **A:** Thanks for your insightful question! We believe that whether to introduce a fine-tuning stage prior to TTA should depend on the alignment between the fine-tuning data and the downstream task, as well as any specific task requirements.
>
> Incorporating fine-tuning is beneficial in the following two scenarios: (1) When the labeled training data and test data **share the same or highly overlapping categories**, fine-tuning can lead to significant gains. For example, fine-tuning on labeled ImageNet data yields noticeable improvements on downstream tasks such as ImageNet-V2 (4-shot ImageNet tuning w/o TTA: 51.5%->55.6% on ImageNet-V2). (2) When **robustness or other specific properties are required**, a supervised fine-tuning stage (e.g., adversarial fine-tuning) can provide a more robust initialization for TTA. In both cases, supervised fine-tuning acts as a strong initialization that enhances TTA performance, and the associated computational cost is generally acceptable.
>
> Conversely, **if the labeled data is not strongly correlated with the test distribution**, supervised fine-tuning tends to generalize poorly (4-shot ImageNet tuning w/o TTA: 55.8%->56.2% on fine-grained dataset). In such scenarios, we recommend applying TTA directly, as it can better adapt to downstream tasks.
>
> We will incorporate the above discussion in our revision to clarify the relationship between training-time and test-time methods.
>
> > Q3: While the paper considers different datasets, they are still limited in diversity, all being image classification dataset.
>
> **A:**  Thank you for highlighting this point. Our current benchmark is indeed focused on classification tasks using vision-language models. This is primarily because most existing TTA algorithms, with the exception of RLCF, are specifically designed for classification problems. Although TTA methods for other tasks (e.g., segmentation) have been proposed, they are relatively recent, limited in number, and often rely on different experimental setups, making it difficult to include them in a unified benchmark alongside classification-focused methods.
>
> We will incorporate a discussion of those methods for other tasks (as shown in the following table) and cite relevant work in the revision. We believe that our benchmark framework can still inspire experimental settings and help guide the development of TTA approaches for a broader range of tasks.
>
> | Method    | TTA paradigm |       Task |
> |------------|-------------|---------------------------|
> | AnoCLIP [1]    | episodic TTA |       Anomaly Localization |
> | T3AL  [2]     | episodic TTA |        Action Localization |
> | DTS-TPT [3]  | episodic TTA |       Activity Recognition |
> | CLIP-DIY [4] | episodic TTA |      Semantic Segmentation |
> | CLIPtrase [5] | episodic TTA |      Semantic Segmentation |
> | VocAda [6]  | episodic TTA |           Object Detection |
> | TTCS   [7] |   online TTA | Medical Image Segmentation |
> | TEST-V [8] |   online TTA |       Video Classification |
>
> [1] Deng et al. "Bootstrap fine-grained vision-language alignment for unified zero-shot anomaly localization." arXiv 2023.
>
> [2] Liberatori et al. "Test-time zero-shot temporal action localization." CVPR 2024.
>
> [3] Yan et al. "DTS-TPT: dual temporal-sync test-time prompt tuning for zero-shot activity recognition." IJCAI 2024.
>
> [4] Wysoczańska et al. "Clip-diy: Clip dense inference yields open-vocabulary semantic segmentation for-free." WACV 2024.
>
> [5] Shao et al. "Explore the potential of clip for training-free open vocabulary semantic segmentation." ECCV 2024.
>
> [6] Liu et al. "Test-time vocabulary adaptation for language-driven object detection." arXiv 2025.
>
> [7] Chen et al. "Test-time medical image segmentation using clip-guided sam adaptation." BIBM 2024.
>
> [8] Yan et al. "TEST-V: test-time support-set tuning for zero-shot video classification." arXiv 2025.
>
> > Q4: It is unclear how big is the distribution shift between pretraining data and TTA data.
>
> **A:** For VLM-based classification problems, it is challenging to quantitatively measure the distribution shift between domains in the same way as traditional domain adaptation (DA) tasks. In conventional DA, the source and target domains typically share the same label space, enabling domain gaps to be assessed through statistical differences between image distributions.
>
> In contrast, VLMs are pretrained on large-scale image-text pairs, while downstream evaluation often involves images with category names. This mismatch in modality and label structure makes direct statistical comparison impractical.
>
> Nevertheless, **the zero-shot performance of a VLM on a given downstream task can still serve as an indirect proxy for task difficulty or the extent of distribution shift between the pretrained model and the task**.
>
> A notable exception arises when the VLM is further finetuned on a dataset (e.g., ImageNet), and the test set shares the same label space (e.g., ImageNet-V2). In such cases, traditional domain adaptation techniques [9,10] can be applied to quantify the distribution shift.
>
> [9] Ben-David et al. "A theory of learning from different domains." Machine learning 2010.
>
> [10] Gretton et al. "A kernel two-sample test." The journal of machine learning research 2012.

---

### Note · Authors · 2025-08-14

We sincerely thank all reviewers and ACs for their thoughtful engagement and valuable feedback. The discussion period was precious, and we are pleased to address questions and concerns from the reviewers. Our work presents **the first systematic benchmark for Test-Time Adaptation (TTA) of Vision-Language Models**, enabling fair and consistent evaluation of diverse algorithms under standardized settings. Through extensive experiments across several datasets, we uncover some observations of existing TTA algorithms, including **limited performance gains, poor collaboration with training-time methods, and reduced model trustworthiness**.

The rebuttal process has further strengthened our submission. We have **expanded our efficiency analysis** by quantifying runtime and memory costs, **clarified the trade-offs** between fine-tuning and direct TTA, and **broadened the benchmark’s scope** by citing some TTA works for tasks beyond classification, such as segmentation, detection, anomaly localization, and video classification. We have also **investigated the impact of prompt quality** through experiments with multiple and more descriptive templates, **extended model coverage** with preliminary results on ViT-base and VisionMamba-base to demonstrate generalization beyond CLIP, and **provided feature space visualizations** to provide qualitative insights into how different adaptation strategies reshape embeddings.

These enhancements significantly deepen the scope and applicability of our work. By providing a rigorous, extensible, and fair evaluation platform, we believe our benchmark will guide future research in making VLMs more robust and efficient in real-world deployment. We thank the reviewers and AC again for their detailed and valuable feedback, which has directly contributed to strengthening this submission.

---

### Decision · Program_Chairs · 2025-09-18

**Decision:**

Accept (poster)

**Comment:**

The article proposes a large-scale benchmark of test-time adaptation for vision-language models, considering 15 methods, 15 datasets, and various backbones. The evaluation tests methods under the same setting, providing insights into their actual gains and the consequences of TTA.

Reviewers provided an initial positive feedback, appreciating the definition of a reproducible evaluation protocol (1qXX, nJYJ, dwCD), the interesting insights (1qXX, nJYJ), depth (4Juw), and the clarity of the work (1qXX, dwCD). They also shared some concerns on the lack of analyses on computational cost (1qXX, nJYJ) and pretraining data (1qXX), and the focus was limited to classification (1qXX, nJYJ, 4Juw). The rebuttal addressed these concerns, with all reviewers providing a positive final rating. Specifically, as TTA for VLMs is often tested on classification, the choice of the benchmark is not a problem, and the supplementary material already contains results on computational complexity.

The AC went through the reviews and the rebuttal and agrees with the reviewers' assessment. The authors are encouraged to include the promised discussion and analyses in the camera-ready.